# OCCAM: Towards Cost-Efficient and Accuracy-Aware Classification Inference

**Dujian Ding, Bicheng Xu, Laks V. S. Lakshmanan**
University of British Columbia
dujian.ding@gmail.com   {bichengx, laks}@cs.ubc.ca

## Abstract

Classification tasks play a fundamental role in various applications, spanning domains such as healthcare, natural language processing and computer vision. With the growing popularity and capacity of machine learning models, people can easily access trained classifiers as a service online or offline. However, model use comes with a cost and classifiers of higher capacity (such as large foundation models) usually incur higher inference costs. To harness the respective strengths of different classifiers, we propose a principled approach, OCCAM, to compute the best classifier assignment strategy over classification queries (termed as the *optimal model portfolio*) so that the aggregated accuracy is maximized, under user-specified cost budgets. Our approach uses an *unbiased* and *low-variance* accuracy estimator and effectively computes the optimal solution by solving an integer linear programming problem. On a variety of real-world datasets, OCCAM achieves $40\%$ cost reduction with little to no accuracy drop.

## 1 Introduction

Classification is a fundamental task with numerous real-world applications, from medical diagnosis (Abhisheka et al., 2024) to natural language processing (Kowsari et al., 2019) and object recognition (Zou et al., 2023). In recent years, a wealth of pre-trained classifiers has become accessible, both online and offline, at varying costs. For instance, Google's Facial Detection service (Google) charges \$1.50 per 1,000 images, providing developers with a reliable solution for face recognition. Simultaneously, a variety of open-source models, such as ResNet (He et al., 2016), Vision Transformer (Dosovitskiy et al., 2020), and Swin Transformer (Liu et al., 2021), have been made available for developers to integrate into their services.

On the one hand, empirical evaluations conducted in (Su et al., 2018) and our independent assessment (see Figure 1a), consistently indicate that smaller/cheaper models tend to exhibit a gap in classification accuracy compared to their larger counterparts. On the other hand, though larger neural models are equipped with higher capacity, they often come with higher costs, e.g., hardware usage and latency, for both training and inference. This can potentially impose an enormous cost on both end users of classification services and the service providers (e.g., Google[1], Amazon[2], and Microsoft[3]).

Confronted with the general trade-off between classification accuracy and inference cost, we advocate a hybrid inference framework which seeks to combine the advantages of both small and large models. Specifically, we study the problem, *how to optimally select and assign models to different classification queries in order to achieve the highest accuracy while adhering to a limited cost budget?* We formally refer to it as the *optimal model portfolio* problem (details in Section 3). Our approach is motivated by the observation that while small classifiers typically have reduced accuracy over the population, they can still agree with large classifiers on certain queries a large proportion of the time, which suggests the existence of a subset of "easy" queries on which even small classifiers can make the right prediction. This is also illustrated in Figure 1b where we plot the frequency with which different classifiers successfully make the right prediction on the same classification queries. Taking

---

[1]https://cloud.google.com/prediction
[2]https://aws.amazon.com/machine-learning
[3]https://studio.azureml.net

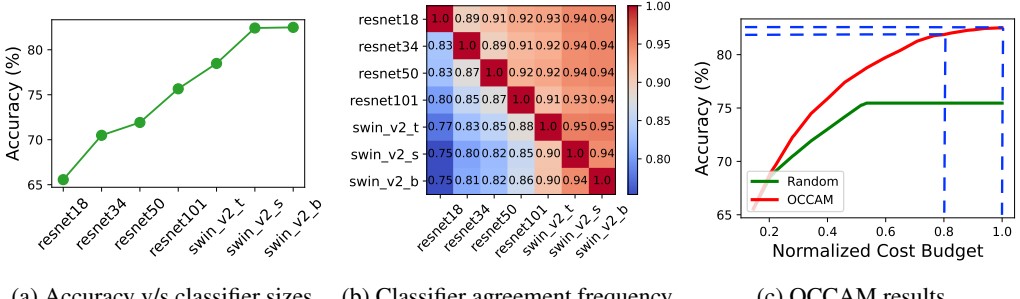

(a) Accuracy v/s classifier sizes.  (b) Classifier agreement frequency.  (c) OCCAM results.

Figure 1: We investigate Tiny ImageNet dataset consisting of 200 classes (see Section 5 for details). We observe that (a) smaller classifiers (e.g., ResNet-18) generally yield lower accuracy, (b) small classifiers can agree with large classifiers at a high frequency (each entry indicates the percentage of queries on which the classifier on the row makes the right prediction and so does the classifier on the column), (c) our approach OCCAM achieves 20% cost reduction with less than 1% accuracy drop.

image classification as an example, a small classifier, ResNet-18 (He et al., 2016), can correctly classify 75% of the images on which the large classifier, SwinV2-B (Liu et al., 2022), makes the right prediction, suggesting that we can replace SwinV2-B with ResNet-18 on these image queries, saving significant inference costs without any accuracy drop (details in Section 5).

With this insight, we propose a principled approach, **O**ptimization with **C**ost **C**onstraints for **A**ccuracy **M**aximization (**OCCAM**), to effectively identify easy queries and assign classifiers to different user queries to maximize the overall classification accuracy subject to the given cost budgets. Previous work (Chen et al., 2022) trains ML models to select classifiers, which requires sophisticated configuration and lacks performance guarantees that are critical in real-world scenarios. In this paper, by leveraging weak assumptions, such as the *well-separation* assumption (Yang et al., 2020), we demonstrate that it is possible to compute optimal model assignments with statistical guarantees. As an example, in previous work (Yang et al., 2020), the image classification task (Singh & Singh, 2020), a critical and well-studied application of classification, has been observed to empirically satisfy the well-separation assumption, which is also corroborated by our own independent evaluation (see Appendix A.1). The intuition is that for well-separated classification problems such as image classification, we can learn robust classifiers that have similar performance on similar queries. By leveraging the specific structure of well-separated classification tasks, we develop an *unbiased* and *low-variance* estimator for classifier test accuracy with *asymptotic* guarantees. For each classification query, we compute its nearest neighbours in pre-computed samples to estimate the test accuracy for each classifier. To our best knowledge, we are the first to open up the black box by developing a white-box accuracy estimator for ML classifiers with statistical guarantees. Next, armed with our classifier accuracy estimator, we compute the optimal classifier assignment strategy over all classification queries (*optimal model portfolio*) subject to a given cost budget by solving an integer linear programming (ILP) problem (see Section 4). As a preview, Figure 1c shows that OCCAM can achieve 20% cost reduction with less than 1% accuracy drop. We show even higher cost reduction with little to no accuracy drop on various real-world datasets in Section 5. Figure 2 depicts the overall pipeline of OCCAM: (1) draw samples and pre-compute the ML classifier accuracy for each sample, (2) for each test query, use its nearest neighbour from all samples to estimate the test accuracy of each classifier, (3) compute the optimal model portfolio w.r.t. cost budgets by solving the ILP problem.

Our main technical contributions are: (1) we formally define the *optimal model portfolio* problem to reduce overall inference costs while maintaining high performance subject to user-specified cost budgets (Section 3); (2) we propose a novel and principled approach, OCCAM, to effectively compute the optimal model portfolio with statistical guarantees under weak assumptions (Section 4); and (3) as an illustration of the usefulness of OCCAM, we conduct extensive experiments on the image classification task given its well-separation property and demonstrate the effectiveness of OCCAM on a variety of real-world datasets (Section 5).

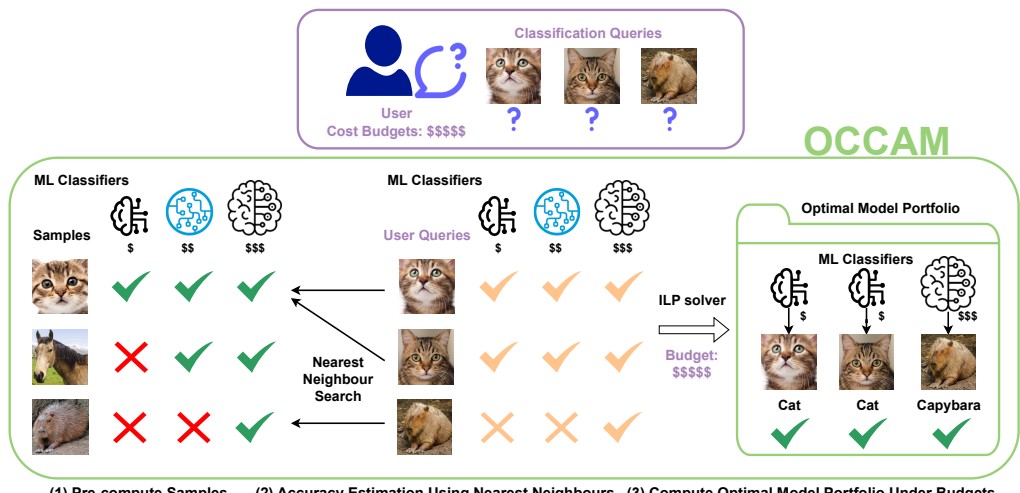

Figure 2: OCCAM: Optimization with Cost Constraints towards Accuracy Maximization.

## 2 RELATED WORK

**Efficient Machine Learning (ML) Inference.** Efficient ML inference is crucial for real-time decision-making in various applications such as autonomous vehicles (Tang et al., 2021), healthcare (Miotto et al., 2018), and fraud detection (Alghofaili et al., 2020). It involves applying a pre-trained ML model to make predictions, where the inference cost is expected to dominate the overall cost incurred by the model. Model compression, which replaces a large model with a smaller model of comparable accuracy, is the most common approach employed for enhancing ML inference efficiency. Common techniques for model compression include model pruning (Hassibi et al., 1993; LeCun et al., 1989; Ding et al.), quantization (Jacob et al., 2018; Vanhoucke et al., 2011), knowledge distillation (Hinton et al., 2015; Urban et al., 2016), and neural architecture search (Elsken et al., 2019; Zoph & Le, 2016). These *static* efficiency optimizations typically lead to a *fixed* model with lower inference cost but also reduced accuracy compared to its larger counterpart, which may not suffice in highly sensitive applications like collision detection (Wang et al., 2021) and prognosis prediction (Zhu et al., 2020). This shortcoming is already evident in the inference platforms discussed in Section 1, highlighting the need for more *dynamic* optimizations to effectively address the diverse demands of users.

**Hybrid ML Inference.** Recent works (Kag et al., 2022; Ding et al., 2022; 2024) have introduced a novel inference paradigm termed hybrid inference, which invokes models of different sizes on different queries, as opposed to employing a single model on all inference queries. The smaller model generally incurs a lower inference cost but also exhibits reduced accuracy compared to the larger model. The key idea is to identify easy inference queries on which the small models are likely to make correct predictions and invoke small models on them when cost budgets are limited, thereby reducing overall inference costs while preserving solution accuracy. By adjusting the cost budgets, users can *dynamically* trade off between accuracy and cost within the same inference setup. Kag et al. (2022); Ding et al. (2022; 2024) consider a simple setting of only one large and one small model and do not allow for explicit cost budget specification, which could be necessary for production scenarios. Chen et al. (2020) studies a setup with multiple ML models and learns an adaptive strategy to generate predictions by calling a *base* model and sometimes an *add-on* model when the base model quality scores are lower than the learned thresholds. However, both base and add-on models are selected in a probabilistic manner and this approach fails to satisfy the user-specified cost budgets deterministically. Chen et al. (2022) studies a similar setup with multiple ML models and allocates cost budgets according to model-based accuracy prediction. This approach requires a separate training phase for the accuracy predictor, which needs a large amount of training data, and provides no guarantee on the prediction quality. Mixture-of-experts (MoE) (Shazeer et al., 2017) also involves dynamically deciding which model (or "expert") processes a given query based on specific criteria. However, MoE often assumes homogenous experts (Fedus et al., 2022) with routing decisions made

by learned gating functions that require significant computational resources to re-train if we want to add/update/delete experts. Unlike previous works, we propose an *unbiased* and *low-variance* accuracy estimator with *asymptotic* guarantees, based on which we present a novel training-free approach, OCCAM, to effectively compute the optimal assignment of diverse classifiers to given queries, under the cost budgets specified by users. It is worth pointing out that OCCAM is a an instance from the Algorithm Selection problem (Rice, 1976) with a specific focus on classification tasks and pre-trained classifiers.

**Image Classification.** Image classification is a fundamental task in computer vision, where given an image, a label needs to be predicted. It serves as an essential building block for many high-level AI tasks, e.g., image captioning (Vinyals et al., 2015) and visual question answering (Antol et al., 2015), where objects need to be first recognized. With the growing capacity of deep learning models, from convolutional neural networks (CNN) (Krizhevsky et al., 2012; Simonyan & Zisserman, 2014) to Transformer architectures (Dosovitskiy et al., 2020; Liu et al., 2021), the classification accuracy on standard image classification benchmarks (Russakovsky et al., 2015) has been greatly improved. In this work, we use the image classification task to illustrate and evaluate our proposed approach OCCAM given its well-separation property (Yang et al., 2020). We utilize both CNN (e.g., ResNet models) and Transformer (e.g., Swin Transformers) image classifiers in our evaluation.

## 3 PROBLEM DEFINITION

Let $\mathcal{X} \subseteq \mathbb{R}^d$ be an instance space (e.g., images) equipped with a metric $dist$: $\mathcal{X} \times \mathcal{X} \to \mathbb{R}^{\geq 0}$, and $[C] = \{1, 2, \cdots, C\}$ be the set of possible labels with $C \geq 2$. Let $\mathcal{X}$ contain $C$ disjoint classes, $\mathcal{X}^{(1)}, \mathcal{X}^{(2)}, \cdots, \mathcal{X}^{(C)}$ where for each $i \in [C]$, all $x \in \mathcal{X}^{(i)}$ have label $i$. Let $f_1, f_2, \cdots, f_M$ be a set of classifiers, with $b_i$ being the cost of a single inference call of $f_i$. Given a query $x \in \mathcal{X}$, each classifier $f_i$ outputs a single label from $[C]$ at the cost $b_i$. We define a *model portfolio* as follows.

**Definition 3.1** (Model Portfolio). Given queries $X \subseteq \mathcal{X}$ to be classified and classifiers $f_1, f_2, \cdots, f_M$, a model portfolio $\mu$ is a mapping $\mu : X \to [M]$ such that *each $x \in X$ is classified by the classifier $f_{\mu(x)}$.*

We assume an oracle classifier $O$: $\mathcal{X} \to [C]$ which outputs the ground truth label $O(x)$ for all queries $x \in X$. Given a finite set of queries $X \subseteq \mathcal{X}$, the *accuracy of a model portfolio $\mu$ on $X$* is the frequency of the ground truth labels correctly predicted by $\mu$, i.e., $Accuracy_X(\mu) = \frac{\sum_{x \in X} \mathbb{1}\{f_{\mu(x)}(x) = O(x)\}}{|X|}$, where $\mathbb{1}\{condition\}$ is an indicator function that outputs 1 iff *condition* is satisfied. Similarly, the *cost of model portfolio $\mu$ on $X$* is the sum of all inference costs incurred by executing $\mu$ on $X$, i.e., $Cost_X(\mu) = \sum_{x \in X} b_{\mu(x)}$. We will use the notation $Accuracy(\mu), Cost(\mu)$ when $X$ is clear from the context. We define our problem as follows.

**Definition 3.2** (Optimal Model Portfolio). Given queries $X \subseteq \mathcal{X}$, a cost budget $B \in \mathbb{R}^+$, and classifiers $f_1, f_2, \cdots, f_M$, find the optimal model portfolio $\mu^*$ such that $Cost(\mu^*) \leq B$ and $Accuracy(\mu^*) \geq Accuracy(\mu)$, for all model portfolios $\mu$ with $Cost(\mu) \leq B$.

## 4 METHODOLOGY

We describe the general framework to solve the *optimal model portfolio* problem in the next sections. Our overall strategy consists of two steps. Firstly, we propose an *unbiased low-variance* estimator for the accuracy of any given model portfolio $\mu$, with *asymptotic guarantees*. Next, we describe how to determine the optimal model portfolio $\mu^*$ by formulating it as an integer linear programming (ILP) problem, subject to user-specified budget constraints. All proofs can be found in Appendix B.

### 4.1 ESTIMATING $Accuracy(\mu)$

Previous work on Hybrid ML (Kag et al., 2022; Chen et al., 2022; Ding et al., 2024) typically relies on training a neural router to predict the accuracy of a given set of classifiers for given user queries, based on which queries are routed to different classifiers. Such a paradigm not only involves a non-trivial training configuration but also lacks estimation guarantees which can be critical in scientific and production settings. We propose a principled approach to estimate the test accuracy of

a given model portfolio for given user queries. By leveraging the specific structure of *well-separated* classification problems like image classification, we propose an *unbiased low-variance* estimator for the test accuracy with *asymptotic* guarantees.

Without loss of generality, we consider the wide class of *soft classifiers* in this study. Given query $x \in \mathcal{X}$, a soft classifier first outputs a distribution over all labels $[C]$, based on which it then makes prediction *at random*. Given a soft classifier $f_i$, we abuse the notation and let $f_i(x)[j]$ denote the likelihood that $f_i$ predicts label $j \in [C]$, that is, $f_i(x)[j] := Pr[f_i(x) = j]$, for query $x \in \mathcal{X}$. Deterministic classifiers (e.g., the oracle $O$) can be seen as a special case of soft classifiers with one-hot distribution over all labels. In practice, from softmax classifiers (e.g., ResNet), soft classifiers can be constructed by simply sampling w.r.t. the probability distribution output by the softmax layer.

Clearly, given a model portfolio $\mu$, $Accuracy(\mu)$ is a random variable due to the random nature of the soft classifiers. The expected accuracy of any given model portfolio $\mu$ is,

$$\mathbb{E}[Accuracy(\mu)] = \mathbb{E}\left[\frac{\sum_{x \in X} \mathbb{1}\{f_{\mu(x)}(x) = O(x)\}}{|X|}\right] = \frac{\sum_{x \in X} \mathbb{E}[\mathbb{1}\{f_{\mu(x)}(x) = O(x)\}]}{|X|} = \frac{\sum_{x \in X} f_{\mu(x)}(x)[O(x)]}{|X|} \quad (1)$$

where the last equality follows from $\mathbb{E}[\mathbb{1}\{f_{\mu(x)}(x) = O(x)\}] = 1 \cdot Pr[f_{\mu(x)}(x) = O(x)] = f_{\mu(x)}(x)[O(x)]$. Note that $f_i(x)[O(x)]$ is the *success probability* that the classifier $f_i$ correctly predicts the ground truth label for query $x$. For brevity, we define $SP_i(x) := f_i(x)[O(x)]$ and rewrite the expected accuracy as $\mathbb{E}[Accuracy(\mu)] = \frac{\sum_{x \in X} SP_{\mu(x)}(x)}{|X|}$.

The exact computation of success probability is intractable since the ground truth of user queries is unknown *a priori*. We propose a novel data-driven approach to estimate it for any classifier and show that our estimator is *unbiased* and *low-variance* with asymptotic guarantees, for *well-separated* classification problems like image classification. Based on this, we develop a principled approach for estimating the expected accuracy of a given model portfolio.

**Definition 4.1** (*r*-separation (Yang et al., 2020))**.** We say a metric space $(\mathcal{X}, dist)$ where $\mathcal{X} = \cup_{i \in [C]} \mathcal{X}^{(i)}$ is *r-separated*, if there exists a constant $r > 0$ such that $dist(\mathcal{X}^{(i)}, \mathcal{X}^{(j)}) \geq r, \forall i \neq j$, where $dist(\mathcal{X}^{(i)}, \mathcal{X}^{(j)}) = \min_{x \in \mathcal{X}^{(i)}, x' \in \mathcal{X}^{(j)}} dist(x, x')$.

In words, in an *r*-separated metric space, there is a constant $r > 0$, such that the distance between instances from different classes is at least $r$. The key observation is that many real-world classification tasks comprise of distinct classes. For instance, images of different categories (e.g., gold fish, bullfrog, etc.) are very unlikely to sharply change their classes under minor image modification. It has been widely observed (Yang et al., 2020) that the classification problem on real-world images empirically satisfies *r*-separation under standard metrics (e.g., $l_\infty$ norm). We also observe similar patterns on a number of standard image datasets (e.g., Tiny ImageNet) and provide more empirical evidence in Appendix A.1. With this observation, we can show that the oracle classifier $O$ is Lipschitz continuous [4] (Eriksson et al., 2004).

**Definition 4.2** (Lipschitz Continuity)**.** Given two metric spaces $(\mathbf{X}, d_{\mathbf{X}})$ and $(\mathbf{Y}, d_{\mathbf{Y}})$ where $d_{\mathbf{X}}$ (resp. $d_{\mathbf{Y}}$) is the metric on the set $\mathbf{X}$ (resp. $\mathbf{Y}$), a function $f: \mathbf{X} \to \mathbf{Y}$ is Lipschitz continuous if there exists a constant $L \geq 0$ s.t.

$$\forall x, x' \in \mathbf{X}: \quad d_{\mathbf{Y}}(f(x), f(x')) \leq L \cdot d_{\mathbf{X}}(x, x') \quad (2)$$

and the smallest $L$ satisfying Equation (2) is called the Lipschitz constant of $f$.

**Lemma 4.3.** *There exists an oracle classifier $O$ which is Lipschitz continuous if the metric space associated with the instances $\mathcal{X}$ is $r$-separated.*

If we further choose the classifiers $f_i$ to be Lipschitz continuous (e.g., MLP (Bartlett et al., 2017), ResNet (Gouk et al., 2021), Lipschitz continuous Transformer (Qi et al., 2023)), we can show that the success probability function $SP_i(x)$ (i.e., the likelihood that a classifier $f_i$ successfully predicts the ground truth label for query $x$) is also Lipschitz continuous.

**Lemma 4.4.** *The success probability function $SP_i(x) = f_i(x)[O(x)]$ is Lipschitz continuous if $f_i(x)$ and $O(x)$ are Lipschitz continuous.*

An important implication of Lemma 4.4 is that, given a classifier, we can estimate its success probability on query $x$ by its success probability on a similar query $x'$. Let $L_i > 0$ denote the Lipschitz

---

[4]The Lipschitz continuity for soft classifiers is defined w.r.t. the output distribution.

constant for $SP_i$. For any $x, x' \in X$, we have the estimation error bounded by $|SP_i(x') - SP_i(x)| \leq L_i \cdot dist(x, x')$, which monotonically decreases as $dist(x, x')$ approaches 0 [5]. In practice, we can pre-compute a labelled sample $S \subset \mathcal{X}$ (e.g., pre-compute classifier outputs on sampled queries from the validation set) and compute $NN_S(x)$, the nearest neighbour of $x$ in $S$, for success probability estimation. We show that the estimator is asymptotically *unbiased*, as sample size increases.

**Lemma 4.5** (Asymptotically Unbiased Estimator). *Given query $x$, a classifier $f_i$, and uniformly sampled $S \subset \mathcal{X}$,*

$$\lim_{s \to \infty} \mathbb{E}[SP_i(NN_S(x))] = SP_i(x) \tag{3}$$

*where $s$ is the sample size and $NN_S(x) := \arg\min_{x' \in S} dist(x, x')$ is the nearest neighbour of $x$ in sample $S$.*

In practice, we draw $K$ i.i.d. samples, $S_1, S_2, \cdots, S_K$, and compute the average sample accuracy $\widehat{SP}_i(x) := \frac{1}{K} \sum_{k=1}^{K} SP_i(NN_{S_k}(x))$ as the estimator of the test accuracy on query $x$, for each classifier $f_i$. It follows from Lemma 4.5 that $\widehat{SP}_i$ is also an asymptotically *unbiased* estimator. We further show below that $\widehat{SP}_i$ is an asymptotically *low-variance* estimator to $SP_i$, as $K$ increases.

**Lemma 4.6** (Asymptotically Low-Variance Estimator). *Given query $x$, a classifier $f_i$, and $K$ i.i.d. uniformly drawn samples $S_1, S_2, \cdots, S_K$ of size $s$, let $\sigma_i^2$ denote the variance of the estimator $\widehat{SP}_i(x)$. We have that $\sigma_i^2$ is asymptotically proportional to $\frac{1}{\sqrt{K}}$ as both $s$ and $K$ increase.*

## 4.2 COMPUTING $\mu^*$ WITH $Accuracy(\mu)$

In the previous section, we show how to estimate the accuracy for a given model portfolio. For each classifier $f_i$ and query $x$, we propose to estimate its success probability $SP_i(x)$ based on similar queries from labelled samples $\widehat{SP}_i(x)$, which can be efficiently pre-computed.

With the estimator in place, we formulate the problem of finding the optimal model portfolio as an integer linear programming (ILP) problem as follows. Given a set of $M$ classifiers $f_1, f_2, \cdots, f_M$, user queries $X = \{x_1, x_2, \cdots, x_N\}$, pre-computed samples $S_1, S_2, \cdots, S_K$, and budget $B \in \mathbb{R}^+$, we have the following ILP problem [6].

$$\max \sum_{i=1}^{M} \sum_{j=1}^{N} \widehat{SP}_i(x_j) \cdot t_{i,j} \quad \text{s.t.} \sum_{i=1}^{M} \sum_{j=1}^{N} b_i \cdot t_{i,j} \leq B \text{ and } \sum_{i=1}^{M} t_{i,j} = 1, \text{ for } j = 1, 2, \cdots, N \tag{4}$$

where $t_{i,j} \in \{0, 1\}$ are boolean variables and $t_{i,j} = 1$ iff the classifier $f_i$ is assigned to query $x_j$. Clearly, the optimal model portfolio $\mu^*$ can be efficiently computed as $\mu^*(x_j) = i$ iff $t_{i,j}^* = 1$, for $i \in [M]$ and $j \in [N]$, where $t_{i,j}^*$ is the optimal solution to the ILP problem above. While ILP problems are NP-hard in general, we can use standard ILP solvers (e.g., HiGHS (Huangfu & Hall, 2018)) to efficiently compute the optimal solution in practice.

The optimization problem aims to maximize the estimated model portfolio accuracy and is subject to the risk of overestimation due to selection bias, especially on large-scale problems. Intuitively, a poor classifier with high-variance estimates can be mistakenly assigned to some queries if its performance on those queries is overestimated. We address this by regularizing the accuracy estimate for each classifier by the corresponding estimator variance. Specifically, we optimize the objective $\sum_{i=1}^{M} \sum_{j=1}^{N} (\widehat{SP}_i(x_j) - \lambda \cdot \sigma_i) \cdot t_{i,j}$ in Equation (4), where $\sigma_i$ is the standard deviation of the estimator $\widehat{SP}_i$. As $\sigma_i$ is unknown *a priori*, we use a validation set to estimate $\sigma_i$ for each classifier $f_i$ and tune $\lambda$ for the highest validation accuracy.

The overall algorithm is shown in Algorithm 1. We first pre-compute the ML classifier accuracy for all samples (line 1-3). Next, for each test query, we use its nearest neighbour from all samples to estimate the test accuracy of each classifier (line 4-6). In line 7, we compute the optimal model portfolio with the accuracy estimation subject to cost budgets by solving the ILP problem (see Equation (4)).

---

[5]We evaluate the nearest neighbour distance and estimation error in Appendices A.2 and A.3

[6]Our problem can be rephrased as "selecting for each query, one item (i.e., ML classifier) from a collection (the set of all classifiers) so as to maximize the total value (accuracy) while adhering to a predefined weight limit (cost budget)", which is a classic multiple choice knapsack problem (MCKP) (Kellerer et al., 2004) and the ILP formulation is the natural choice.

---

**Algorithm 1:** OCCAM Algorithm.

---

**Input:** test query batch $X$; ML classifiers $f_1, f_2, \cdots, f_M$ and costs $c_1, c_2, \cdots, c_M$; query
      samples $S_1, S_2, \ldots, S_k$; user cost budget $B$.
**Output:** optimal model portfolio $\mu^* : X \to [M]$.

```
// Pre-compute the ML classifier accuracy for all samples
```
1 **for** $s \in S_1 \cup S_2 \cup \cdots \cup S_k$ **do**
2    **for** $i \in 1, 2, \cdots, M$ **do**
3       $SampleAccuracy[s, i] = Accuracy(f_i(s), ground\_truth(s))$

```
// Use samples to estimate the accuracy for each test query
```
4 **for** $x \in X$ **do**
5    **for** $i \in 1, 2, \cdots, M$ **do**
6       $AccuracyEstimation[x, i] =$
          $Mean(\{SampleAccuracy[NearestNeighbour(x, S_j), i], \text{for } 1 \le j \le k\})$

```
// Compute the optimal model portfolio with the accuracy
   estimation subject to the cost budget (see Equation (4))
```
7 $\mu^* = ILPSolver(AccuracyEstimation, c_1, c_2, \cdots, c_M, B)$
8 **return** $\mu^*$

---

## 5 EVALUATION

### 5.1 EVALUATION SETUP

**Task.** We consider the image classification task provided with its well-separation property (Yang et al., 2020): given an image, predict a class label from a set of predefined class categories. We assume that each image has a unique ground-truth class label.

**Datasets.** We consider 4 widely studied datasets for image classification: CIFAR-10 (10 classes) (Krizhevsky et al., 2009), CIFAR-100 (100 classes) (Krizhevsky et al., 2009), Tiny ImageNet (200 classes) (CS231n), and ImageNet-1K (1000 classes) (Russakovsky et al., 2015). Details of those datasets are in Appendix C.1.

**Models.** We consider a total of 7 classifiers: ResNet-[18, 34, 50, 101] (He et al., 2016)[7] and SwinV2-[T, S, B] (Liu et al., 2022)[8] Among these classifiers, ResNet-18 is the smallest (in terms of number of model parameters and training/inference time) and thus has the least capacity, while SwinV2-B is the largest and with the highest accuracy in general (see Figure 1a). We take the classifiers pre-trained on the ImageNet-1K dataset (Russakovsky et al., 2015). We directly use the pre-trained models on ImageNet-1K, while on other datasets, we freeze everything but train only the last layer from scratch. Model training details are in Appendix C.2. All experiments are conducted with one NVIDIA V100 GPU of 32GB GPU RAM. Codes are available in https://github.com/DujianDing/OCCAM.git.

**Inference Cost.** The absolute costs of running a model may be expressed using a variety of metrics, including FLOPs, latency, dollars, etc. While FLOPs is an important metric that has the advantage of being hardware independent, it has been found to not correlate well with wall-clock latency, energy consumption, and dollar costs, which are of more practical interest to end users (Dao et al., 2022). In practice, dollar costs usually highly correlate with inference latency on GPUs. In our work, we define the cost of model inference in USD. We approximate the inference cost of computation by taking the cost per hour ($3.06) of the Azure Machine Learning (AML) NC6s v3 instance (AzureML, 2024), as summarized in Table 1. The AML NC6s v3 instance contains a single V100 GPU and is commonly used for deep learning. Since CPU resources are significantly cheaper than GPU (e.g., D2s v3 instance, equipped with two 2 CPUs and no GPU, costs $0.096 per hour (AzureML, 2024)) and all methods studied in this work typically finish in several CPU-seconds, incurring negligible expenses, we ignore the costs incurred by CPU in our comparison. In addition, since larger models typically have higher accuracy as well as higher costs (see Figure 1a), a practically interesting setting is to study how to deliver high quality answers with reduced costs in comparison to solely using the

---

[7]Numbers in bracket indicate the model's layer number.
[8]Letters in bracket indicate the Swin Transformer V2's size. T/S/B means tiny/small/base.

Table 1: Model costs on the image classification task. Latency and prices are measured for 10,000 queries. Normalized cost is the fraction of the price w.r.t. SwinV2-B.

| Models | Latency (s) | Prices ($) | **Normalized Cost** |
|---|---|---|---|
| ResNet-18 | 88.9 | 0.076 | **0.15** |
| ResNet-34 | 135.9 | 0.116 | **0.22** |
| ResNet-50 | 174.5 | 0.148 | **0.29** |
| ResNet-101 | 317.4 | 0.270 | **0.52** |
| SwinV2-T | 326.4 | 0.277 | **0.53** |
| SwinV2-S | 600.7 | 0.511 | **0.98** |
| SwinV2-B | 610.6 | 0.519 | **1** |

largest model (e.g., SwinV2-B). Normalized cost directly indicates the percentage cost saved and has been widely adopted in previous works (Ding et al., 2024; Kag et al., 2022), following which we report all results in terms of the normalized cost of each classifier.

**ILP Solver.** While our approach is agnostic to the choice of the ILP solver, we choose the high-performance ILP solver, HiGHS (Huangfu & Hall, 2018) to solve the problem in Equation (4), given its well-demonstrated efficiency and effectiveness on public benchmarks (Gleixner et al., 2021). In a nutshell, HiGHS solves ILP problems with branch-and-cut algorithms (Fischetti & Monaci, 2020) and stops whenever the gap between the current solution and the global optimum is small enough (e.g., 1e-6).

**Baselines.** We compare our approach with three baselines: *single best*, *random*, and FrugalMCT (Chen et al., 2022). *Single best* always chooses the strongest (i.e., most expensive) model for a given cost budget. *Random* estimates classifier accuracy with random guesses (i.e., uniform samples from $[0, 1]$) and solves the problem in Equation (4) with the same ILP solver as ours. FrugalMCT (Chen et al., 2022) is a recent work which selects the best ML models for given user budgets in an online setting, using model-based accuracy estimation. Following the same setting in (Chen et al., 2022), we train random forest regressors on top of the model-extracted features (e.g., ResNet-18 features), as the accuracy predictor. The predicted accuracy is used in Equation (4), which is solved by the same ILP solver as ours to make a fair comparison in terms of solution quality and costs.

**Our Method.** We evaluate OCCAM (see Section 4) under various metrics (i.e., $l_1$, $l_2$, and $l_\infty$ norms) and cost budgets. We consider images represented by model-based embeddings. Specifically, we extract the image feature [9] of the query image and all the validation images. The costs incurred by feature extraction are deducted from the user budget $B$ before we compute the optimal model portfolio. We report the test accuracy under different cost budgets for OCCAM and all baselines in Section 5.2 (Figure 3 and Table 2), validate that OCCAM is cost-aware and indeed selecting the most *profitable* ML models to deliver high accuracy solutions in Section 5.3 (Figure 4a), demonstrate the effectiveness of OCCAM with limited samples and various $\lambda$ values in Section 5.4 (Figures 4b and 4c), investigate the nearest neighbour distance in Appendix A.2, show that the estimation error of our accuracy estimator quickly decreases as the sample size increases in Appendix A.3, test the generalizability of OCCAM with different feature extractors in Appendix A.4, provide more performance results under different metrics (see Appendix A.5) and more classifiers (see Appendix A.6), show how to effectively choose $K$ and $s$ in Appendix A.7, perform overhead and upper bound performance analysis of OCCAM in Appendices A.8 and A.9, and examine the classifier complementarity in Appendix A.10.

For simplicity, unless otherwise stated, we report OCCAM performance using ResNet-18 features and $l_\infty$ metric with $K = 40$ for all datasets ($s = 500$ for CIFAR10, CIFAR100, and $s = 1000$ for Tiny ImageNet, ImageNet-1K). We choose $\lambda = 100$ for ImageNet-1K and $\lambda = 5$ for all other datasets because ImageNet-1K contains a high variety of image classes (1000 classes) that leads to relatively high estimation errors and requires more regularization penalty via large $\lambda$ values.

## 5.2 PERFORMANCE RESULTS

We investigate the test accuracy achieved by OCCAM and all baselines under different cost budgets and depict the results in Figure 3. We can see that by trading little to no accuracy drop, OCCAM achieves significant cost savings and outperforms all baselines across a majority of experiment

---

[9]The image feature is the last layer output of a ML model (e.g., ResNet-18) trained on the target dataset, given an input image.

Table 2: Cost reduction v.s. accuracy drop by OCCAM and baselines. Cost reduction and accuracy drops are computed w.r.t. using the single largest model (i.e., SwinV2-B) for all queries. For example, on Tiny ImageNet, using SwinV2-B to classify all $10,000$ test images achieves an accuracy of $82.5\%$ and incurs a total cost of $\$0.519$ (we take it as the normalized cost 1, see Table 1). A $10\%$ cost reduction equals a cost budget of $\$0.467$ (i.e., a normalized cost 0.9), under which we evaluate the achieved accuracy of OCCAM and all baselines and report the relative accuracy drops.

| Cost Reduction (%) | Accuracy Drop (%) | | | | | | | |
|---|---|---|---|---|---|---|---|---|
| | CIFAR10 | | | | CIFAR100 | | | |
| | Single Best | Rand | Frugal -MCT | OCCAM | Single Best | Rand | Frugal -MCT | OCCAM |
| 10 | 2.22 | 2.86 | 0.97 | **0.56** | 3.18 | 3.29 | 0.52 | **0.34** |
| 20 | 2.22 | 2.86 | 1.13 | **0.50** | 3.18 | 3.29 | 0.79 | **0.36** |
| 40 | 2.22 | 2.86 | 1.22 | **0.51** | 3.18 | 3.29 | 1.98 | **0.62** |
| Cost Reduction (%) | Tiny-ImageNet-200 | | | | ImageNet-1K | | | |
| | Single Best | Rand | Frugal -MCT | OCCAM | Single Best | Rand | Frugal -MCT | OCCAM |
| 10 | 4.01 | 7.03 | 0.86 | **0.17** | 2.53 | 5.98 | 0.59 | **0.51** |
| 20 | 4.01 | 7.03 | 1.49 | **0.61** | 2.53 | 5.98 | 1.12 | **1.05** |
| 40 | 4.01 | 7.03 | 3.88 | **2.75** | 2.53 | 5.98 | 2.35 | **2.24** |

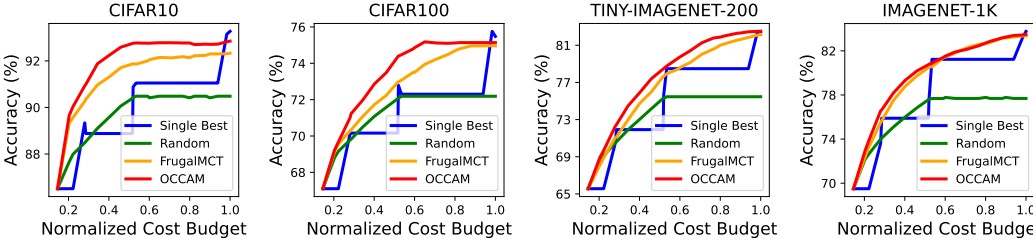

Figure 3: Accuracy-cost tradeoffs achieved by OCCAM and baselines, for different cost budgets.

settings. Results on cost reduction vs accuracy drop for all approaches are summarized in Table 2. On *easy classification task* (CIFAR-10 of 10 classes), OCCAM consistently outperforms all baselines by achieving $40\%$ cost reduction with up to $0.56\%$ accuracy drop. Cost reduction and accuracy drop are computed w.r.t. using the strongest model (i.e., SwinV2-B) for all queries. On *moderate classification task* (CIFAR-100 of 100 classes), OCCAM outperforms all baselines by trading up to $0.62\%$ accuracy drop for $40\%$ cost reduction. On *hard classification task* (Tiny ImageNet of 200 classes), OCCAM significantly outperforms all three baselines with at least $0.5\%$ higher accuracy. Notably, on aggressive cost regimes (e.g., $40\%$ cost reduction), the achieved accuracy of OCCAM is $1.1\%$ higher than FrugalMCT, $4.3\%$ higher than *random*, and $1.3\%$ higher than *single best*. On *the most challenging classification task* (ImageNet-1K of 1000 classes), OCCAM still consistently outperforms all three baselines with higher accuracy at all cost budget levels. We believe that the above results demonstrate the generalized effectiveness of OCCAM in achieving non-trivial cost reduction for a small accuracy drop on classification tasks of different difficulty levels.

## 5.3 VALIDATION RESULTS

We validate that OCCAM is functioning as intended, that is, it does select *small-yet-profitable* classifiers when budgets are limited and gradually switches to *large-but-accurate* classifiers as cost budgets increase. In Figure 4a we plot the model usage for each classifier under different cost budgets on the Tiny ImageNet dataset. From the figure, it can be seen that when cost budgets are restricted, OCCAM mainly chooses ResNet-18 to resolve queries given its low costs and good accuracy (as seen in Table 1 and Figure 1a). As budgets increase, OCCAM gradually switches to SwinV2-S and SwinV2-B, given their predominantly high accuracy ($82\%$ as seen in Figure 1a).

## 5.4 STABILITY ANALYSIS

OCCAM pre-computes $K$ labelled samples of size $s$ to estimate the test accuracy at inference time and computes the optimal model portfolio by solving Equation (4) with regularized accuracy estimation,

$\widehat{SP}_i(x) - \lambda \cdot \sigma_i$ (see Section 4.2). We investigate OCCAM performance with different total sample sizes ($K \cdot s$) by setting $s = 1000$ and changing $K$ from $10$ to $40$ (see Figure 4b), and examine the stability of OCCAM to the choice of $\lambda$ (see Figure 4c). We report results on Tiny ImageNet dataset.

In Figure 4b, we plot the achieved accuracy of OCCAM under different total sample sizes ($K \cdot s$) and normalized cost budgets ($B$). We also report FrugalMCT performance using a maximum of $40,000$ sampled images to train its accuracy predictor. With budget $B = 0.8$ (i.e., $20\%$ cost reduction), OCCAM

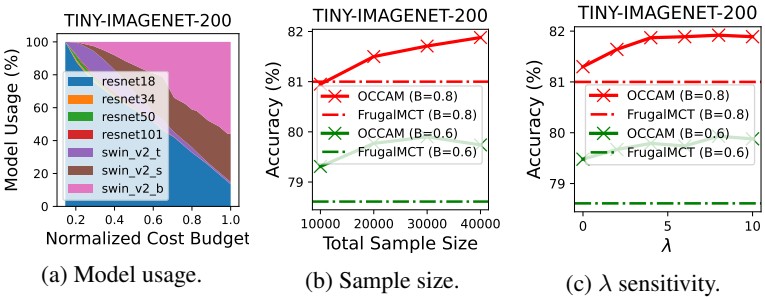

(a) Model usage.  (b) Sample size.  (c) $\lambda$ sensitivity.

Figure 4: (a) Model usage breakdown by OCCAM under different cost budgets, (b) OCCAM accuracy with different sample sizes, and (c) OCCAM performance with different $\lambda$ choices

CAM achieves comparable performance to FrugalMCT at $25\%$ samples and continues to outperform FrugalMCT as the total sample size increases. With budget $B = 0.6$ (i.e., $40\%$ cost reduction), OCCAM outperforms FrugalMCT by $0.7\%$ higher accuracy with only $25\%$ samples and achieves up to $1.3\%$ higher accuracy as the total sample size increases, which demonstrates the sustained effectiveness of OCCAM even with limited samples.

In Figure 4c, we plot the achieved accuracy of OCCAM with different $\lambda$ choices and cost budgets ($B$). Compared to FrugalMCT, OCCAM achieves up to $1.3\%$ higher accuracy when budget $B = 0.6$, and $0.8\%$ higher accuracy when $B = 0.8$. Notably, OCCAM outperforms FrugalMCT under all settings even when $\lambda = 0$ and always achieves higher accuracy when $\lambda > 0$, which justifies the preserved effectiveness of OCCAM even when $\lambda$ is not well-tuned.

## 6   LIMITATIONS

**OCCAM assumes well-separated tasks.** OCCAM assumes that the classification task is *well separated* (see Section 4.1), meaning intuitively that instances (e.g., images) of the problem should not sharply change their class labels under minor modification. To adapt OCCAM to other classification tasks such as sentiment analysis (Medhat et al., 2014), the challenge is how to choose the most suitable numeric representation so that the separation property is preserved.

**OCCAM assumes IID data.** The validity of our theoretical guarantees relies on the assumption that underlying data is independent and identically distributed (IID). In our evaluation, we uniformly sample training and validation splits from the same population to ensure the IID assumption. It is an interesting question how to adapt our approach to accommodate data drifts or out-of-distribution data.

**OCCAM optimizes the average accuracy.** In sensitive applications such as healthcare or autonomous driving, optimizing the average accuracy may not be adequate. Instead, optimizing the min accuracy is more appropriate. We can achieve this by changing the ILP objective (Equation (4)) to maximize the min accuracy of all queries, that is, $\min\{\sum_{i=1}^{M} \widehat{SP}_i(x_j) * t_{i,j}, \text{for } 1 \leq j \leq N\}$.

## 7   CONCLUSION

Motivated by the need to optimize the classifier assignment to different classification queries with pre-defined cost budgets, we have formulated the *optimal model portfolio* problem and proposed a principled approach, **O**ptimization with **C**ost **C**onstraints for **A**ccuracy **M**aximization (**OCCAM**), to effectively deliver high accuracy solutions. We present an *unbiased* and *low-variance* estimator for classifier test accuracy with *asymptotic* guarantees, and compute an optimal classifier assignment with novel regularization techniques mitigating overestimation risks. Our experimental results on a variety of real-world classification datasets show that we can achieve up to $40\%$ cost reduction with no significant drop in classification accuracy.

ACKNOWLEDGMENTS

The authors would like to thank Yuxi Feng and Chenyang Tao for helpful discussions. This work is supported in part by the Institute for Computing, Information and Cognitive Systems (ICICS) at UBC. Research by the first and last authors was supported by a grant from the Natural Sciences and Engineering Research Council of Canada (NSERC), Grant Number RGPIN-2020-05408.

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

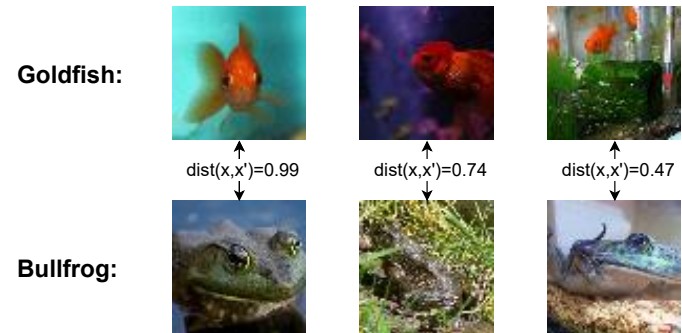

Figure 5: Intuitive example of well separated images from Tiny ImageNet under $l_\infty$ distance metric using ResNet-18 features. Images from different classes (e.g., "goldfish" and "bullfrog") are typically well separated by a non-zero distance.

## A    ADDITIONAL EXPERIMENTS

### A.1    REAL IMAGE DATASETS ARE WELL SEPARATED.

In (Yang et al., 2020), authors have shown that many real image classification tasks comprise of separated classes in RGB-valued space. In this section, we provide further empirical evidence to show that real image datasets (e.g., Tiny ImageNet) are well separated (see Definition 4.1) in different feature spaces under various metrics (Figures 5 and 6).

In Figure 5, we provide an intuitive example to illustrate that images from different classes (e.g., "goldfish" and "bullfrog") are typically well separated by a non-zero distance. In Figure 6, we investigate the distance distribution for images of different classes from Tiny ImageNet (200 classes). We observe that images of different classes are typically far from each other by a non-zero distance under different metrics (e.g., $l_1$, $l_2$, and $l_\infty$) in different feature spaces (e.g., image features extracted by ResNet-18, ResNet-50, and SwinV2-T).

In addition, we note that real image datasets are subject to little to no label noises (Zhu et al., 2024). For example, on Tiny ImageNet, we investigate $40,000$ images from the training split and only find 4 duplicate images of different class labels. We also consider more standard image datasets (see Section 5.1). It turns out that CIFAR-10 contains no label noise, CIFAR-100 contains 3 duplicate images of different class labels (out of $20,000$ images), and the noise frequency on ImageNet-1K is 8 out of $40,000$ images. Our observation suggests that standard image datasets are quite clean (aligned with the observation in (Yang et al., 2020)) that justifies the adoption of well-separation assumption.

### A.2    NEAREST NEIGHBOUR DISTANCE APPROACHES 0 AS SAMPLE SIZE INCREASES.

In this section, we conduct experiments to investigate the changes of nearest neighbour distance $(dist(x, NN_S(x)))$ as sample size $(s)$ increases. We report results using different feature extractors (ResNet-18, ResNet-50, and SwinV2-T) as well as different metrics ($l_1$, $l_2$, and $l_\infty$) on the validation split of Tiny ImageNet dataset (Figure 7).

It can be clearly seen in Figure 7 that the distance to the sampled nearest neighbour quickly approaches 0 as sample size increases. This could be attributable to the fact that we are sampling from real images. With properly pre-trained feature extractors, the possible image embeddings could be restricted to a subspace rather than pervade the whole high-dimensional space, which can significantly reduce the required number of samples and give us meaningfully small distances to the sampled nearest neighbours.

Another interesting observation is that, in all investigated feature space, $l_\infty$ always provides the smallest nearest neighbour distance with different sample sizes, followed by $l_2$ and $l_1$. Such distinction mainly results from the fact that we use normalized image features where each dimension of the feature vector $x$ is between 0 and 1, that is, $0 \le x[i] \le 1$ for any $x[i] \in x$. Consequently,

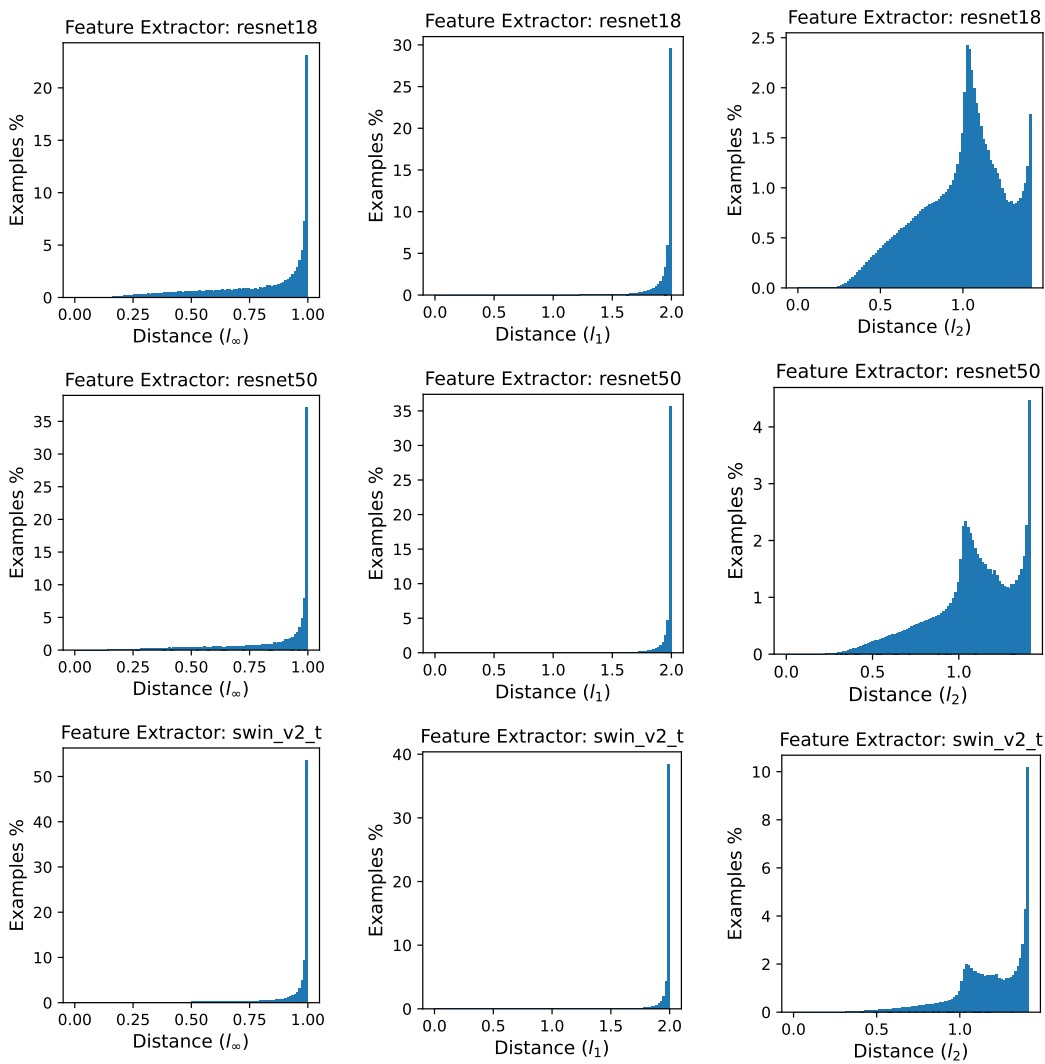

Figure 6: Distance distribution between images of different classes from Tiny Imagenet. We consider image representation derived by different feature extractors (ResNet-18, ResNet-50, and SwinV2-T) as well as different metrics ($l_1$, $l_2$, and $l_\infty$). Images of different classes are typically far from each other by non-zero distances.

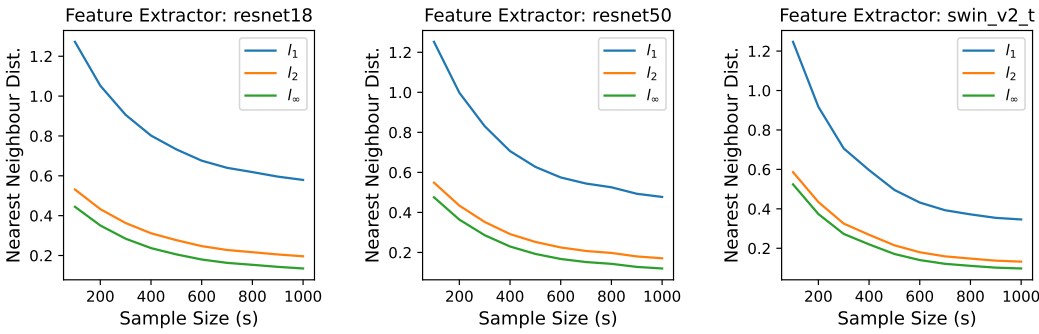

Figure 7: Nearest neighbour distance quickly approaches 0 as the sample size increases using different image feature extractors (ResNet-18, ResNet-50, and SwinV2-T) and metrics ($l_1$, $l_2$, and $l_\infty$).

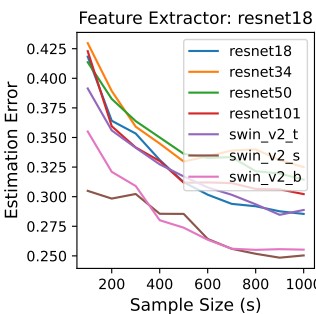 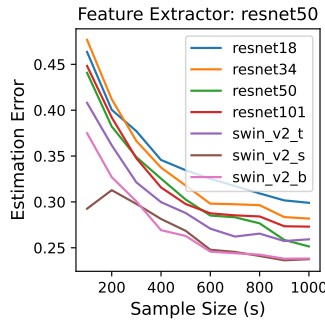 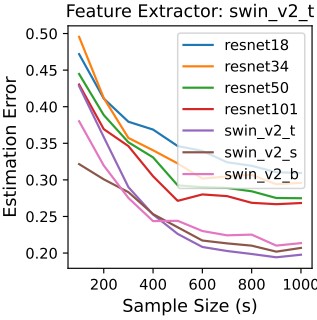

Figure 8: Estimation error for each ML classifier quickly decreases as the sample size increases using different image feature extractors (ResNet-18, ResNet-50, and SwinV2-T) under $l_\infty$ metrics.

Table 3: Cost reduction v.s. accuracy drop by baselines and OCCAM using different feature extractors (ResNet-18, ResNet-50, and SwinV2-T) and $l_\infty$ distance metric. Cost reduction and accuracy drops are computed w.r.t. using the single largest model (i.e., SwinV2-B) for all queries.

| | Accuracy Drop (%) | | | | | | | |
|---|---|---|---|---|---|---|---|---|
| Cost | Tiny-ImageNet-200 | | | | | | | |
| Reduction (%) | Single Best | Rand | FrugalMCT (ResNet-18) | FrugalMCT (ResNet-50) | FrugalMCT (SwinV2-T) | OCCAM (ResNet-18) | OCCAM (ResNet-50) | OCCAM (SwinV2-T) |
| 10 | 4.01 | 7.03 | 0.86 | 0.84 | 1.18 | 0.48 | 0.40 | **0.29** |
| 20 | 4.01 | 7.03 | 1.49 | 1.45 | 1.60 | 1.02 | 0.74 | **0.58** |
| 40 | 4.01 | 7.03 | 3.88 | 4.12 | 3.22 | 3.24 | **2.56** | 2.81 |

we have the inequality that the $l_\infty(x) = \max\{|x[i]| \, | \, x[i] \in x\} \le l_2(x) = \sqrt{\sum_{x[i] \in x} |x[i]|^2} \le l_1(x) = \sum_{x[i] \in x} |x[i]|$. Recall that the OCCAM employs the classifier accuracy estimator which is asymptotically unbiased as nearest neighbour distance approaches 0. The above observation suggests that $l_\infty$ is likely to provide smaller nearest neighbour distance and reduce the estimation error that leads to higher overall performance, especially in scenarios when sampling is expensive or labelled data is scarce.

### A.3 ESTIMATION ERROR DECREASES AS SAMPLE SIZE INCREASES.

In this section, we investigate the estimation error (difference between real classifier accuracy and our estimator results) for different ML classifiers, using different feature extractors (ResNet-18, ResNet-50, and SwinV2-T). For brevity, on Tiny ImageNet, we report the estimation error in the accuracy of all 7 classifiers (ResNet-[18, 34, 50, 101], and SwinV2-[T, S, B]), under $l_\infty$ metric (Figure 8). The patterns are similar with other metrics and feature extractors.

It is clear from Figure 8 that the estimation error of our accuracy estimator continues to decrease for all ML classifiers as the sample size increases, which demonstrates the effectiveness our accuracy estimator design (see Section 4.1).

### A.4 GENERALIZING TO DIFFERENT FEATURE EXTRACTORS

We further report the performance of OCCAM with different feature extractors (ResNet-18, ResNet-50, and SwinV2-T), on TinyImageNet. As in illustrated in Section 5.1, the costs incurred by feature extraction are "deducted from the user budget before we compute the optimal model portfolio". Results are summarized in Table 3. It can be seen that OCCAM outperforms all baselines on all experimental settings, which demonstrates the effectiveness and generalizability of OCCAM with different feature extractors.

Table 4: Cost reduction v.s. accuracy drop by OCCAM and baselines using ResNet-18 features and $l_1$ distance metric. Cost reduction and accuracy drops are computed w.r.t. using the single largest model (i.e., SwinV2-B) for all queries.

| | Accuracy Drop (%) | | | | | | | |
|---|---|---|---|---|---|---|---|---|
| Cost | CIFAR10 | | | | CIFAR100 | | | |
| Reduction (%) | Single Best | Rand | Frugal -MCT | OCCAM | Single Best | Rand | Frugal -MCT | OCCAM |
| 10 | 2.22 | 2.86 | 0.97 | **0.38** | 3.18 | 3.29 | 0.52 | **0.50** |
| 20 | 2.22 | 2.86 | 1.13 | **0.38** | 3.18 | 3.29 | 0.79 | **0.50** |
| 40 | 2.22 | 2.86 | 1.22 | **0.37** | 3.18 | 3.29 | 1.98 | **0.99** |
| Cost | Tiny-ImageNet-200 | | | | ImageNet-1K | | | |
| Reduction (%) | Single Best | Rand | Frugal -MCT | OCCAM | Single Best | Rand | Frugal -MCT | OCCAM |
| 10 | 4.01 | 7.03 | 0.86 | **0.48** | 2.53 | 5.98 | **0.59** | 0.86 |
| 20 | 4.01 | 7.03 | 1.49 | **1.02** | 2.53 | 5.98 | **1.12** | 1.51 |
| 40 | 4.01 | 7.03 | 3.88 | **3.24** | 2.53 | 5.98 | **2.35** | 3.32 |

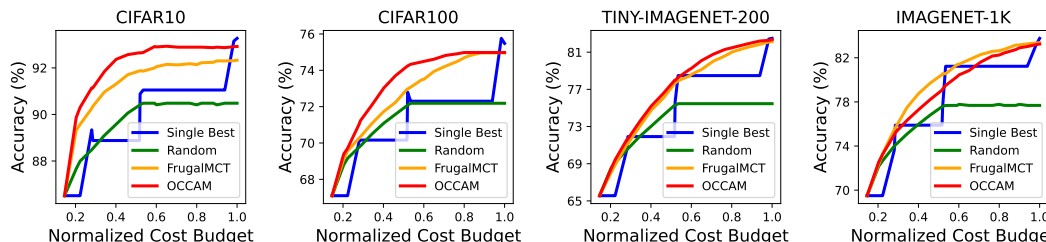

Figure 9: Accuracy-cost tradeoffs achieved by OCCAM and baselines using $l_1$ metric and ResNet-18 features, for different cost budgets.

## A.5    MORE OCCAM PERFORMANCE RESULTS.

In this section, we provide more OCCAM performance results using $l_1$ and $l_2$ norm metrics, as shown in Figures 9 and 10. Qualitative comparison results are summarized in Tables 4 and 5, which resemble our analysis in Section 5.2. Typically, by trading little to no performance drop, OCCAM can achieve significant cost reduction and outperform all baselines across a majority of experiment settings.

However, we also note that FrugalMCT can sometimes outperform OCCAM on ImageNet-1K using $l_1$ and $l_2$ metrics, while OCCAM outperforms FrugalMCT across all experiment settings using $l_\infty$ metric (see Section 5.2). This could be explained by the fact that $l_1$ and $l_2$ metrics are likely to provide higher nearest neighbour distance than $l_\infty$ metric (see Appendix A.2) that implicitly increases OCCAM estimator error and leads to reduced overall performance, especially when the classification task is challenging and labelled data is scarce. Provided that, in practice, we would recommend applying OCCAM with $l_\infty$ to achieve significant cost reduction with little to no performance drop (see Section 5.2).

## A.6    LARGE-SCALE EXPERIMENTS WITH 101 PRE-TRAINED CLASSIFIERS.

In this section, we investigate the capacity of OCCAM on ImageNet-1K dataset with 101 pre-trained models for image classification (e.g., DenseNet (Huang et al., 2017), ResNet (He et al., 2016) to VGG (Simonyan & Zisserman, 2014), ViT (Dosovitskiy et al., 2020) to Swin Transformer V2 (Liu et al., 2022)) that are available on Pytorch Hub [10]. We plot the validation accuracy and costs of the 101 pre-trained models in Figure 11. It can be clearly seen that a majority of pre-trained models are Pareto-dominated by others, that is, there exist models which are more accurate yet cheaper. The subset of models that are not Pareto-dominated by any other models are termed as *Pareto-efficient* models. In practice, we can always replace a Pareto-dominated model with a corresponding Pareto-efficient

---

[10]https://pytorch.org/vision/main/models.html

Table 5: Cost reduction v.s. accuracy drop by OCCAM and baselines using ResNet-18 features and $l_2$ distance metric. Cost reduction and accuracy drops are computed w.r.t. using the single largest model (i.e., SwinV2-B) for all queries.

| Cost Reduction (%) | Accuracy Drop (%) | | | | | | | |
|---|---|---|---|---|---|---|---|---|
| | CIFAR10 | | | | CIFAR100 | | | |
| | Single Best | Rand | Frugal -MCT | OCCAM | Single Best | Rand | Frugal -MCT | OCCAM |
| 10 | 2.22 | 2.86 | 0.97 | **0.24** | 3.18 | 3.29 | 0.52 | **0.34** |
| 20 | 2.22 | 2.86 | 1.13 | **0.25** | 3.18 | 3.29 | 0.79 | **0.40** |
| 40 | 2.22 | 2.86 | 1.22 | **0.27** | 3.18 | 3.29 | 1.98 | **0.71** |
| Cost Reduction (%) | Tiny-ImageNet-200 | | | | ImageNet-1K | | | |
| | Single Best | Rand | Frugal -MCT | OCCAM | Single Best | Rand | Frugal -MCT | OCCAM |
| 10 | 4.01 | 7.03 | 0.86 | **0.21** | 2.53 | 5.98 | **0.59** | 1.06 |
| 20 | 4.01 | 7.03 | 1.49 | **0.81** | 2.53 | 5.98 | **1.12** | 1.65 |
| 40 | 4.01 | 7.03 | 3.88 | **2.75** | 2.53 | 5.98 | **2.35** | 3.10 |

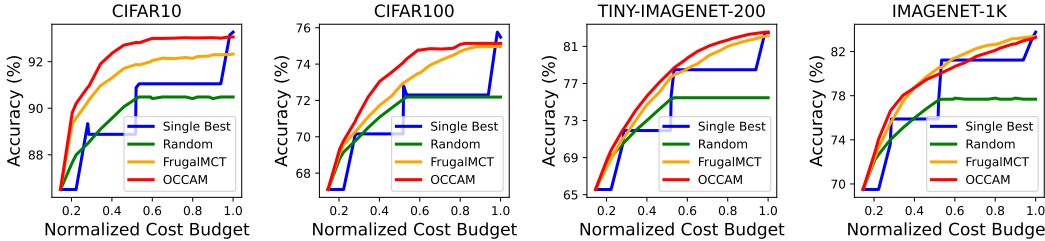

Figure 10: Accuracy-cost tradeoffs achieved by OCCAM and baselines using $l_2$ metric and ResNet-18 features, for different cost budgets.

model to improve accuracy and reduce costs simultaneously, which serve as the *core set* of models to choose from. Specifically, there are 12 Pareto-efficient models from the 101 pre-trained models as summarized in in Table 6. In our evaluation, we examine OCCAM with the 12 Pareto-efficient models and compare it with baselines of access to all 101 pre-trained models. Note that FrugalMCT does not finish after running 5 hours and we exclude it from the comparison. Results are shown in Figure 12 and Table 7. OCCAM consistently outperforms all baselines by achieving less than 1% accuracy drop at various cost reduction rates, while the accuracy drop for Random is up to 7.05%, and 15.09% for Single Best. It is worth noting that, Single Best, as defined in 5.1, always chooses the most expensive model for a given cost budget. As shown in Figure 11, in practice, more expensive models are not necessarily of higher accuracy. The performance of Single Best oscillates heavily due to the existence of expensive-and-inaccurate models, which suggests that blindly using hundreds of classifiers in practice may not eventually help the overall performance.

## A.7 Effects of K and s on OCCAM Performance.

In this section, we examine the effects of $K$ and $s$ on the performance of OCCAM (see Figure 13). As shown by Lemmas 4.5 and 4.6, our estimator is asymptotically unbiased and low-variance as both $K$ (number of samples) and $s$ (sample size) increase, which leads to better overall performance. We demonstrate this by increasing $K$ from 10 to 40 while keeping $s$ fixed (1000), and increasing $s$ from 400 to 1,000 while keeping $K$ fixed (40), as illustrated in Figures 13a and 13b. We observe that the performance of OCCAM consistently dominates that of FrugalMCT and improves as both $K$ and $s$ increase. In our evaluation, we pre-compute a held-out dataset (e.g., for TinyImageNet, we uniformly sample 40,000 images from the training split) from which we draw $K$ samples of size $s$. With the maximal number of samples bounded (for TinyImageNet, $K * s \leq 40000$), there is a trade-off between increasing $K$ and having larger $s$. As shown in Figure 13c, though a larger $K$ typically leads to better accuracy, it also limits the maximal value that $s$ can take ($s \leq 40000/K$),

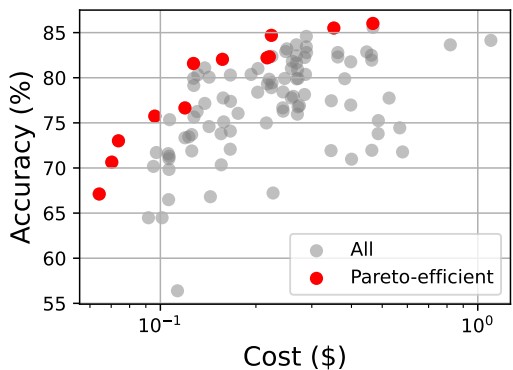

Figure 11: Accuracy v.s. cost for the 101 pre-trained classifiers on ImageNet-1K from Pytorch Hub.

Table 6: Accuracy and costs of the 12 Pareto-efficient models on the image classification task. Latency and prices are measured for 10,000 queries. Normalized cost is the fraction of the price w.r.t. RegNet_Y_128GF (the most expensive model).

| Models | Accuracy (%) | Latency (s) | Prices ($) | Normalized Cost |
|---|---|---|---|---|
| ResNet-18 | 67.1 | 75.5 | 0.064 | **0.14** |
| VGG19 | 70.6 | 82.7 | 0.070 | **0.15** |
| VGG19_BN | 73.0 | 86.8 | 0.074 | **0.16** |
| MNASNet1_3 | 75.7 | 113.0 | 0.096 | **0.21** |
| RegNet_X_800MF | 76.7 | 140.6 | 0.120 | **0.26** |
| ViT_B_16 | 81.6 | 149.6 | 0.127 | **0.27** |
| ConvNeXt_Tiny | 82.0 | 184.6 | 0.157 | **0.34** |
| RegNet_X_16GF | 82.2 | 255.1 | 0.217 | **0.46** |
| RegNet_X_32GF | 82.3 | 259.3 | 0.220 | **0.47** |
| ViT_L_16 | 84.7 | 263.2 | 0.224 | **0.48** |
| ViT_H_14 | 85.5 | 414.1 | 0.352 | **0.75** |
| RegNet_Y_128GF | 86.0 | 549.7 | 0.467 | **1.00** |

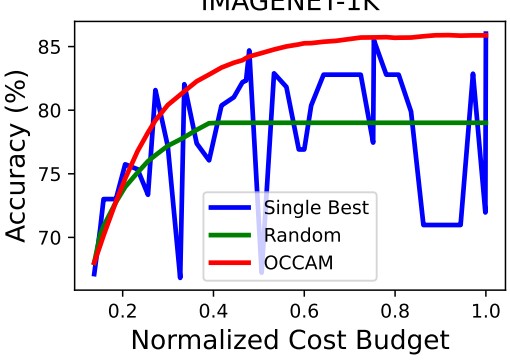

Figure 12: Accuracy-cost tradeoffs by OCCAM and baselines using $l_\infty$ metric and ResNet-18 features with the 101 pre-trained classifiers on ImageNet-1K.

Table 7: Cost reduction v.s. accuracy drop by baselines and OCCAM using $l_\infty$ metric and ResNet-18 features with the 101 pre-trained classifiers on ImageNet-1K. Cost reduction and accuracy drops are computed w.r.t. using the most expensive model (i.e., RegNet_Y_128GF) for all queries. FrugalMCT does not finish after 5 hours and is excluded from the comparison.

| | Accuracy Drop (%) | | | |
|---|---|---|---|---|
| Cost Reduction (%) | ImageNet-1K | | | |
| | Single Best | Rand | FrugalMCT | OCCAM |
| 10 | 15.09 | 7.05 | N/A | 0.17 |
| 20 | 3.27 | 7.05 | N/A | 0.37 |
| 40 | 9.15 | 7.05 | N/A | 0.80 |

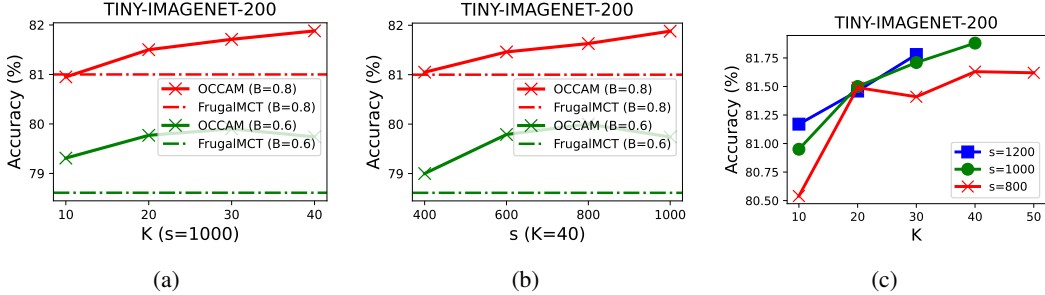

(a)           (b)           (c)

Figure 13: OCCAM performance improves as (a) the number of samples $K$ and (b) the sample size $s$ increase. (c) In practice, when the total sample size is fixed (e.g., 40,000 for Tiny ImageNet), there is a trade-off between increasing $K$ and having larger $s$.

which results in sub-optimal performance. We empirically choose $K$ and $s$ that give us the best overall accuracy across different datasets, as discussed in Section 5.1.

## A.8 OVERHEAD ANALYSIS

In this section, we investigate the overheads incurred by OCCAM, which mainly result from nearest neighbour search and the use of the ILP solver, on the Tiny ImageNet dataset.

With 10,000 test images and 40,000 total samples, the nearest neighbor search takes 8.68 seconds to return all results, up to two orders of magnitude smaller than the model inference time (see Table 1), which leads to negligible overheads. Such efficiency can be attributed to the linear time complexity of nearest neighbor search w.r.t. the total sample sizes. Specifically, let $N$ denote the number of test images, $s$ denote the sample size, $K$ denote the number of samples, and $d$ denote the image representation dimension; then the time complexity of nearest neighbor search is $O(N * s * K * d)$.

In our evaluation, we adopt HiGHS (Huangfu & Hall, 2018) as our ILP solver given its well-demonstrated efficiency and effectiveness on public benchmarks (Gleixner et al., 2021). With 10,000 test images and 7 classifiers (equivalently an ILP instance with 70,000 variables and constraints, see Equation (4)), the HiGHS ILP solvers takes 15.1 seconds to return the optimal assignment. It is worth noting that the latency overhead of ILP solver is only fractional to using the smallest model (ResNet-18 takes 88.9s to process 10,000 test images, see Table 1), and even less than 2.5% of always using the largest model (SwinV2-B takes 610.6s to process 10,000 test images, see Table 1), which demonstrates the overhead incurred by ILP solver is negligible.

We note that there are other ILP solvers which are even more efficient than HiGHS. For example, Gurobi [11] is up to 20X faster than HiGHS on large scale MILP instances with up to 640K variables and constraints (Sun et al., 2023), which can be used as the ILP solver if the problem scale increases further.

---

[11]https://www.gurobi.com/

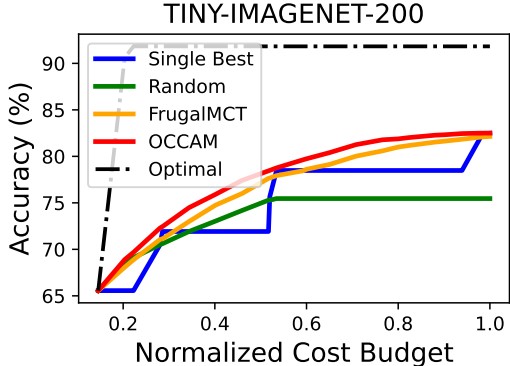

Figure 14: Accuracy-cost tradeoffs by OCCAM and baselines using $l_\infty$ metric and ResNet-18 features. Optimal is computed using the true classification likelihood instead of estimation.

Overall, the latency overheads incurred by OCCAM are negligible. Since OCCAM only requires CPU support to compute the optimal model assignment, the monetary overheads are significantly small as well.

### A.9 UPPER BOUND PERFORMANCE OF OCCAM.

In this section, we investigate the optimal performance that can be reached with true classification likelihood (no estimation errors) on the Tiny ImageNet dataset (see Figure 14). Note that, the underlying method of OCCAM and the Optimal is exactly the same. The performance difference between OCCAM and Optimal mainly comes from the estimation error and hints at a huge room for further improvements which we will explore in our future work.

### A.10 COMPLEMENTARITY BETWEEN DIFFERENT CLASSIFIERS.

In this section, we examine the complementary behaviour between different classifiers, that is, different classifiers may suit the best for different subset of test queries. On the Tiny ImageNet dataset, we plot the frequency that one classifier makes the right prediction while the other fails (see Figure 15). For example, on 5% of test queries, ResNet-18 is able to correctly classify the labels on which SwinV2-B fails, while on the other 24% of test queries, SwinV2-B is better than ResNet-18 in terms of making right predictions. The complementary nature of different classifiers implies the possibility of developing a model assignment strategy that effectively outperforms the single best classifier such as the Optimal approach shown in Figure 14. It can be seen that the Optimal performance significantly outperforms the Single Best baseline and suggests a huge room for further improvements by reducing the accuracy estimation errors.

## B PROOFS

In this section, we provide proofs to Lemmas 4.3 to 4.6.

**Proof to Lemma 4.3**

*Proof.* The proof is straightforward. Without loss of generality, we consider the $l_1$ metric and assume $(\mathcal{X}, l_1)$ is a $r$-separated metric space. For brevity, we abuse the notation and let $O(x)$ denote the one-hot output distribution over all labels. For any $x, x' \in \mathcal{X}$, if $x$ and $x'$ belong to the same class, then $\|O(x) - O(x')\|_1 = 0 \leq \frac{2}{r} \cdot \|x - x'\|_1$; otherwise, $\|O(x) - O(x')\|_1 = 2 \leq \frac{2}{r} \cdot \|x - x'\|_1$. The Lipschitiz constant for $O$ is $\frac{2}{r}$. $\square$

**Proof to Lemma 4.4**

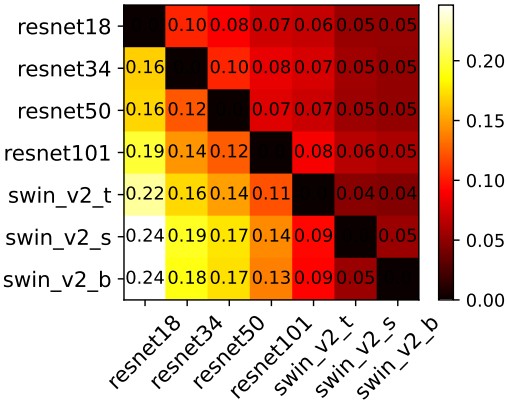

Figure 15: Classifier complementarity. Each entry indicates the percentage of queries on which the classifier on the row makes the right prediction while the classifier on the column fails.

*Proof.* Similarly, without loss of generality, we consider the $l_1$ metric and let $f_i(x), O(x)$ denote the output distribution over all labels. Let $L_i$ and $L_O$ denote the Lipschitz constants for $f_i(x)$ and $O(x)$ respectively. For any $x, x' \in \mathcal{X}$, if $x$ and $x'$ belong to the same class, then $\|SP_i(x) - SP_i(x')\|_1 = |f_i(x)[O(x)] - f_i(x')[O(x')]| \le \|f_i(x) - f_i(x')\|_1 \le L_i \cdot \|x - x'\|_1$; otherwise, $\|SP_i(x) - SP_i(x')\|_1 = |f_i(x)[O(x)] - f_i(x')[O(x')]| \le 1 = \frac{1}{2}\|O(x) - O(x')\|_1 \le \frac{L_O}{2} \cdot \|x - x'\|_1$. The Lipschitiz constant for $SP_i(x)$ is $\max\{L_i, \frac{L_O}{2}\}$. $\qquad\square$

**Proof to Lemma 4.5**

*Proof.* The proof leverages the fact that, as the sample size increases, the expected distance between $x$ and its nearest neighbour monotonically decreases. Letting $L_i$ denote the Lipschitz constant of $SP_i$, we have the estimation error $|\mathbb{E}[SP_i(NN_S(x))] - SP_i(x)| = \mathbb{E}[|SP_i(NN_S(x)) - SP_i(x)|] \le L_i \cdot \mathbb{E}[dist(NN_S(x), x)]$, which approaches 0 as $\mathbb{E}[dist(NN_S(x), x)]$ decreases. $\qquad\square$

**Proof to Lemma 4.6**

*Proof.* Lemma 4.5 shows that each $SP_i(NN_{S_k}(x))$ is an unbiased estimator of $SP_i(x)$, $1 \le k \le K$, as $s$ approaches infinity. Let $\sigma_i'^2$ denote the variance of $SP_i(NN_{S_k}(x))$ for each $k$. By the Central Limit Theorem, the distribution of the estimator $\frac{1}{K}\sum_{k=1}^{K} SP_i(NN_{S_k}(x))$ approaches a normal distribution with variance $\frac{\sigma_i'^2}{\sqrt{K}}$ (Chang et al., 2024). $\qquad\square$

## C   EXPERIMENT DETAILS

### C.1   DATASETS

**CIFAR-10**[12]. CIFAR-10 (Krizhevsky et al., 2009) contains $60,000$ images of resolution $32 \times 32$, evenly divided into 10 classes, where $50,000$ images are for training and $10,000$ images are for testing. We randomly sample $20,000$ images from the training set as our validation set, and we use the remaining $30,000$ images to train our models.

**CIFAR-100**[13]. Same as CIFAR-10, CIFAR-100 (Krizhevsky et al., 2009) has $50,000$ training and $10,000$ testing images. But they are evenly separated into 100 classes. We randomly sample $20,000$ training images as our validation set.

---

[12]https://www.cs.toronto.edu/~kriz/cifar.html
[13]https://www.cs.toronto.edu/~kriz/cifar.html

**Tiny ImageNet**[14]. Tiny ImageNet (CS231n) is a subset of the ImageNet-1K dataset (Russakovsky et al., 2015). It covers 200 class labels and all images are in resolution $64 \times 64$. It includes $100,000$ training, $10,000$ validation, and $10,000$ testing images. The given test split does not have ground-truth labels, thus we discard this set and use the validation split as our testing data. We randomly sample $40,000$ training images as the validation data and use the remaining $60,000$ ones to train the models.

**ImageNet-1K**[15]. We use the image classification dataset in the ImageNet Large Scale Visual Recognition Challenge (ILSVRC) 2012 (Russakovsky et al., 2015). This dataset contains 1,281,167 training, 50,000 validation, and 100,000 testing images, covering 1,000 classes. Images are of various resolutions. Since the models we use are pre-trained on this dataset, we do not train the last linear layer of the models. The given test split comes without ground-truth labels; thus we use the validation split to evaluate our method and baselines. Among the 50,000 validation images, we randomly select 10,000 of them as our testing data and the remaining ones are treated as the validation data.

## C.2 MODELS

We use ResNet (He et al., 2016) and Swin Transformer V2 (SwinV2) (Liu et al., 2022) models on the image classification task because they are popular models for the task and many of their pre-trained weights on the ImageNet-1K dataset (Russakovsky et al., 2015) are available online[16], where reasonable performance are achieved. On CIFAR-10, CIFAR-100, and Tiny ImageNet, we freeze everything of the pre-trained models but only train the last linear layer of each model from scratch. The output dimension of the last layer is set to be the same as the number of image classes on the test dataset. We implement the soft classifier (see Section 4.1) by sampling w.r.t. the probability distribution output by the softmax layer, i.e., $Pr[f_i(x) = j] = \frac{exp(z_j/\tau)}{\sum_k exp(z_k/\tau)}$, where $z_k$ is the logit for $k \in [C]$ and $\tau$ is the hyper-parameter *temperature* controlling the randomness of predictions. We choose a small $\tau$ (1e-3) to reduce the variance in predictions. At test time, to obtain consistent results, all classifiers make predictions by outputting the most likely class labels (i.e., $\arg\max_j f_i(x)[j]$), equivalently to having soft classifiers with $\tau \to 0$. For all seven models, we use the Adam optimizer (Kingma & Ba, 2015) with $\beta_1 = 0.9$ and $\beta_2 = 0.999$, constant learning rate 0.00001, and a batch size of 500 for training. Models are trained till convergence.

---

[14]http://cs231n.stanford.edu/tiny-imagenet-200.zip

[15]https://image-net.org/download.php

[16]For example, the pre-trained models we use are from https://pytorch.org/vision/stable/models.html. Specifically, the pre-trained weights we use are as follows.

- ResNet-18: `ResNet18_Weights.IMAGENET1K_V1`
- ResNet-34: `ResNet34_Weights.IMAGENET1K_V1`
- ResNet-52: `ResNet50_Weights.IMAGENET1K_V1`
- ResNet-101: `ResNet101_Weights.IMAGENET1K_V1`
- SwinV2-T: `Swin_V2_T_Weights.IMAGENET1K_V1`
- SwinV2-S: `Swin_V2_S_Weights.IMAGENET1K_V1`
- SwinV2-B: `Swin_V2_B_Weights.IMAGENET1K_V1`

