# OpenReview forum: "OCCAM: Towards Cost-Efficient and Accuracy-Aware Classification Inference"
_ICLR.cc/2025/Conference — ICLR 2025 Poster_

### Official Review · Reviewer_iJX4 · 2024-11-01

**Soundness:** 3
**Presentation:** 3
**Contribution:** 2
**Rating:** 8
**Confidence:** 3

**Summary:**

The authors present a method for cost-efficient classifier selection at inference time.
The proposed method uses an estimator of classifier accuracy, which is based on the assumption of well-separated classification instances (images) and utilizes samples of classification accuracy from the classifiers.
Given such an estimator, the construction of an optimal model portfolio is stated as an Integer Linear Programming problem.
The method is evaluated using established image classifiers and standard benchmark data.

**Strengths:**

- The proposed method is well-motivated and of great practical relevance for users of image classification services.

- The method sound under assumptions of well-separated instances and the findings are supported both empirically and theoretically

**Weaknesses:**

- The proposed method relies on precomputed samples of the classifiers under consideration. These are assumed to be given without any cost. However, this is typically not the case. In order to be able to use it in practice, these samples need to be collected which results in expenditure. A good contribution would be to analyse the following question:
Given a user-specified budget, how many samples should be acquired in order to maximize accuracy while adhering to the budget with overall cost (samples for accuracy estimation + classification instances).

- Additionally, the use of the nearest-neighbor based accuracy estimator as well as the ILP solver are assumed to incur no cost, which does not hold in practice.

- The considered problem setting is an instance of the (per-instance) Algorithm Selection (AS) problem (Rice 76). That is, given an instance of an algorithmic problem domain and a set of algorithms suitable to solve said instance, select the algorithm that optimizes a performance metric. For the submitted study, the problem domain is image classification, the problem instances are images, the algorithms are image classifiers and the performnace metric is a multi-criteria with cost and accuracy. There is a substantial corpus of literature concerned with AS and also the idea of using algorithm portfolios has been examined. Although the manuscript considers a very specific instantiation of this problem, I think a reference should be given.

Minor remarks:
- p 1. line 37: "On the other (hand)"

Literature:
Rice, J. R. (1976). The algorithm selection problem. In Advances in computers (Vol. 15, pp. 65-118). Elsevier.

**Questions:**

- Is it possible to have a strong (the strongest) baseline by solving the ILP not with an estimator of accuracy but the true classification likelihoods, as an upper bound of achievable performance given a predefined budget?

- To me it seems like the proposed method quite naturally translated into an online setting. It particularly seems to resemble a contextual multi-armed bandit problem, in which each classifier is an arm and the context is given by the classification instances. The reward would be multi-criteria containing costs as well as performance. Would such an extension to online learning make sense?

- The work considers an order of image classifiers, from cheap and less accurate to costly and more accurate. While the classifiers certainly can be ordered wrt accuracy, is there not also a kind of performance complimentarity, i.e. one classifier is better at classifying certain images while another classifier is better suited for other images? I think this may be another strength of OCCAM. Even in (hypothetical) settings in which classifiers have the same cost and the same accuracy on average, OCCAM may be able to identify the better classifer on a per-instance level, effectively outperforming the single best classifier. I am not sure whether these phenomena are present for the considered classifiers and data, however, the possibility should be discussed.

---

> ### Author Response · Authors · 2024-11-23
> **To Reviewer iJX4 (Part I)**
>
> We thank you for your careful review and address your concerns below.
>
> **Q1: The proposed method relies on precomputed samples of the classifiers under consideration. These are assumed to be given without any cost. However, this is typically not the case. In order to be able to use it in practice, these samples need to be collected which results in expenditure. A good contribution would be to analyse the following question: Given a user-specified budget, how many samples should be acquired in order to maximize accuracy while adhering to the budget with overall cost (samples for accuracy estimation + classification instances).**
>
> A1: Thank you for this insightful comment! It is worth noting that all samples are only pre-computed once and the pre-sampling cost can be amortized to nearly zero over the long run. For example, the pre-sampling cost by applying all classifiers on the 40,000 sample images incurs \\$7.7 according to Table 1 (Page 8). The amortized cost quickly decreases to \\$0.00077 / query after 10,000 test queries, and monotonically gets smaller with a factor 1 / # of test queries. Also, we would like to point out that our approach is also sample efficient (see Figure 4b, Page 10) where OCCAM is able to match or outperform the previous SOTA method with only 25% pre-computed samples, which further alleviates the concerns over pre-sampling cost.
>
> On the other hand, we do appreciate the idea of the holistic problem formulation by considering both pre-sampling costs and inference costs under a uniform cost budget, which is essentially meaningful when data drifts are present and we need to adaptively draw samples to accommodate the new data distribution along query processing. We will leave it to our future work.
>
> **Q2: Additionally, the use of the nearest-neighbor based accuracy estimator as well as the ILP solver are assumed to incur no cost, which does not hold in practice.**
>
> A2: Thank you for the great comment.
>
> Nearest neighbor search: On Tiny ImageNet, we investigated the overheads incurred by nearest neighbor search and the use of the ILP solver. With 10,000 test images and 40,000 total samples, the nearest neighbor search takes 8.68 seconds to return all results, up to two orders of magnitude smaller than the model inference time (see Table 1, Page 8), which leads to negligible overheads. Such efficiency can be attributed to the linear time complexity of nearest neighbor search w.r.t. the total sample sizes. Specifically, let N denote the number of test images, s denote the sample size, K denote the number of samples, and d denote the image representation dimension; then the time complexity of nearest neighbor search is O(N\*s\*K\*d).
>
> ILP solver: In our evaluation, we adopt HiGHS [1] as our ILP solver given its well-demonstrated efficiency and effectiveness on public benchmarks [2]. With 10,000 test images and 7 classifiers (equivalently an ILP instance with 70,000 variables and constraints, see E.q. 4 in Page 6), the HiGHS ILP solvers takes 15.1 seconds to return the optimal assignment. It is worth noting that the latency overhead of ILP solver is only a fraction of the time of using the smallest model (ResNet-18 takes 88.9s to process 10,000 test images, see Table 1, Page 8), and actually less than 2.5% of always using the largest model (SwinV2-B takes 610.6s to process 10,000 test images, see Table 1, Page 8), which demonstrates the overhead incurred by ILP solver is negligible.
>
> We note that there are other ILP solvers which are even more efficient than HiGHS. For example, Gurobi [3] is up to 20X faster than HiGHS on large scale MILP instances with up to 640K variables and constraints [4], which can be used as the ILP solver if the problem scale increases further.
>
> Overall, the latency overheads incurred by our method are negligible. Since our method only requires CPU support to compute the optimal model assignment, the monetary overheads are significantly small as well.

---

> ### Author Response · Authors · 2024-11-23
> **To Reviewer iJX4 (Part II)**
>
> **Q3: The considered problem setting is an instance of the (per-instance) Algorithm Selection (AS) problem (Rice 76). That is, given an instance of an algorithmic problem domain and a set of algorithms suitable to solve said instance, select the algorithm that optimizes a performance metric. For the submitted study, the problem domain is image classification, the problem instances are images, the algorithms are image classifiers and the performnace metric is a multi-criteria with cost and accuracy. There is a substantial corpus of literature concerned with AS and also the idea of using algorithm portfolios has been examined. Although the manuscript considers a very specific instantiation of this problem, I think a reference should be given.**
>
> A3: Thank you for this great comment and for the valuable reference. As correctly pointed out in the comment, the well-studied Algorithm Selection (AS) problem [5] shares similarity to OCCAM on a high level, while OCCAM has a focus on image classification and pre-trained classifiers, and leverages the specific problem structure (well-separation structure, see Sec 4.1, Page 5) and classifier property (Lipschitz Continuity, see Sec 4.1, Page 5) to effectively compute the optimal model assignment under given cost budget. We have included the reference to the AS problem literature in our revision and also discussed the relationship with OCCAM (see Sec 2, Page 3 in our revision).
>
> **Q4: p 1. line 37: "On the other (hand)"**
>
> A4: Thank you for the comment. We have fixed it in our revision.
>
> **Q5: Is it possible to have a strong (the strongest) baseline by solving the ILP not with an estimator of accuracy but the true classification likelihoods, as an upper bound of achievable performance given a predefined budget?**
>
>
> A5: Thank you for the comment. We have investigated the optimal performance (Optimal) that can be reached with true classification likelihood as illustrated in our revised manuscript (see Fig 14, Page 21 in our revision). Note that, the underlying method of OCCAM and the Optimal is exactly the same. The performance difference between OCCAM and Optimal mainly comes from the estimation error and hints at a considerable room for further improvements in estimation error, which we will explore in our future work.
>
> **Q6: To me it seems like the proposed method quite naturally translated into an online setting. It particularly seems to resemble a contextual multi-armed bandit problem, in which each classifier is an arm and the context is given by the classification instances. The reward would be multi-criteria containing costs as well as performance. Would such an extension to online learning make sense?**
>
> A6: Thank you for this excellent comment. The extension to online settings makes perfect sense. OCCAM assumes that a reasonable sample can be pre-computed, while in an online setting such as MAB we need to calibrate our estimation by adaptively triggering different models in real-time based on the historical data. It is indeed an intriguing extension to be explored in the future.

---

> ### Author Response · Authors · 2024-11-23
> **To Reviewer iJX4 (Part III)**
>
> **Q7: The work considers an order of image classifiers, from cheap and less accurate to costly and more accurate. While the classifiers certainly can be ordered wrt accuracy, is there not also a kind of performance complimentarity, i.e. one classifier is better at classifying certain images while another classifier is better suited for other images? I think this may be another strength of OCCAM. Even in (hypothetical) settings in which classifiers have the same cost and the same accuracy on average, OCCAM may be able to identify the better classifer on a per-instance level, effectively outperforming the single best classifier. I am not sure whether these phenomena are present for the considered classifiers and data, however, the possibility should be discussed.**
>
>
> A7: Thank you for this really insightful comment. The complementary performance phenomenon is indeed present in our evaluation, as illustrated in Figure 15, Page 22 in our revision, where each entry indicates the percentage of queries on which the classifier on the row makes the right prediction while the classifier on the column fails. For example, on 5% of test queries, ResNet-18 is able to correctly classify the labels on which SwinV2-B fails, while on the other 24% of test queries, SwinV2-B is better than ResNet-18 in terms of making right predictions. As correctly pointed out by the reviewer, the complementary nature of different classifiers implies a model assignment strategy that effectively outperforms the single best classifier, as indicated in Figure 14, Page 21 in our revision where we plot the upper bound of OCCAM (Optimal) assuming the accuracy estimator is perfect. It can be seen that the upper bound performance significantly outperforms the single best baseline and hints at a considerable room for further improvements in estimation error, which we will explore in our future work.
>
>
> Thank you for your time and consideration. We sincerely hope that you find our responses convincing and would consider increasing your rating.
>
>
>
> References:
> [1] https://highs.dev/
> [2] Gleixner, Ambros, et al. "MIPLIB 2017: data-driven compilation of the 6th mixed-integer programming library." Mathematical Programming Computation 13.3 (2021): 443-490.
> [3] https://www.gurobi.com/
> [4] MindOpt Adapter for CPLEX Benchmarking Performance Analysis, 2024
> [5] Rice, J. R. (1976). The algorithm selection problem. In Advances in computers (Vol. 15, pp. 65-118). Elsevier.

---

> > ### Comment · Reviewer_iJX4 · 2024-11-26
> >
> > I want to thank the authors for the effort and care taken in revising the manuscript and clarifying the questions in the response. I have adjusted my score.

---

> > > ### Author Response · Authors · 2024-11-27
> > > **Thank you Reviewer iJX4**
> > >
> > > We highly appreciate your recognition and helpful reviews that enabled us to greatly improve the quality of the paper!

---

### Official Review · Reviewer_K8Qp · 2024-11-01

**Soundness:** 3
**Presentation:** 3
**Contribution:** 3
**Rating:** 6
**Confidence:** 4

**Summary:**

The authors propose a method for choosing a good set of classifiers that try to maximize classification accuracy while maintaining some cost constraints. Their method is based on estimating the expected accuracies of each classifier in the portfolio using nearest neighbor in a suitable embedding space, and then apply integer programming to find out the best classifier for each test example. Experiments show their method beats several state-of-art methods when tested on several image classification benchmarks.

**Strengths:**

- This problem of choosing a good set of classifiers to maximize accuracies under computational budget constraints is a very relevant and practical problem given the rising costs of running neural-network-based classifiers.

- Empirically the method performs very well under various cost reductions when compared to other algorithms such as Frugal-MCT and single best, having higher accuracies at the same cost reduction levels.

- The authors also provide theoretical justifications for their accuracy estimates of different classifiers for a new test sample, under the assumption of Lipschitz continuity and well-separatedness.

**Weaknesses:**

- Choosing the hyperparameter \lambda for different datasets seems difficult. It is set to 100 for Imagenet-1K and 5 for other datasets, which is a large range. This can impact the practical performance of the method.

- The results for unbiasedness and low-variance in Lemma 4.5 and 4.6 are asymptotic. In practice since we are training neural networks for embeddings, the underlying metric space and nearest neighbor function DEPENDS on the training set. For example, the data can be r-separated on the training set since the neural network embedding is trained on it, but not so on a separate validation set. If the samples S_1, ..., S_k in Section 4.2 comes from the training set, the estimates for accuracy can be biased. If they come from a separate validation set then we need a fairly well-represented validation set to estimate the accuracies, which can be a limitation of the method.

- There is a big gap of costs between SwinV2-S and SwinV2-T with no intermediate models. This makes the 10%, 20%, and 40% cost reduction in Table 2 all use the same model for single best and the results for 10% and 20% cost reduction weak for single best since there is no model with intermediate costs available.

**Questions:**

- Why are the results for Random the same for all cost reduction levels in Table 2 and Table 4, if it solves the same ILP problem as in Equation 4?

---

> ### Author Response · Authors · 2024-11-23
> **To Reviewer K8Qp**
>
> We thank you for your careful review and address your concerns below.
>
> **Q1: Choosing the hyperparameter \lambda for different datasets seems difficult. It is set to 100 for Imagenet-1K and 5 for other datasets, which is a large range. This can impact the practical performance of the method.**
>
> A1: Thank you for the comment. As discussed in Lines 421-423, a high variety of image classes (1000 classes) typically leads to relatively high estimation errors and requires more regularization penalty via large \lambda values. It is worth noting that OCCAM continuously outperforms the previous SOTA method by achieving higher accuracy even if \lambda is under-tuned (see Figure 4c, Page 10). In our evaluation, we choose the \lambda value giving the best performance on a held-out set and apply it to the test queries.
>
> **Q2: The results for unbiasedness and low-variance in Lemma 4.5 and 4.6 are asymptotic. In practice since we are training neural networks for embeddings, the underlying metric space and nearest neighbor function DEPENDS on the training set. For example, the data can be r-separated on the training set since the neural network embedding is trained on it, but not so on a separate validation set. If the samples S_1, ..., S_k in Section 4.2 comes from the training set, the estimates for accuracy can be biased. If they come from a separate validation set then we need a fairly well-represented validation set to estimate the accuracies, which can be a limitation of the method.**
>
> A2: Thank you for this insightful comment. As pointed out in this comment, the validity of our theoretical guarantees relies on the assumption that underlying data is independent and identically distributed (IID). In our evaluation, we uniformly sample training and validation splits from the same population to ensure the IID assumption holds. It is an interesting question how to adapt our approach to accommodate data drifts or out-of-distribution data. We have added it to Sec 6, Page 10 in our revision.
>
> **Q3: There is a big gap of costs between SwinV2-S and SwinV2-T with no intermediate models. This makes the 10%, 20%, and 40% cost reduction in Table 2 all use the same model for single best and the results for 10% and 20% cost reduction weak for single best since there is no model with intermediate costs available.**
>
> A3: Thank you for this valuable observation. In our evaluation, we consider 7 mainstream image classifiers (ResNet-[18,34,50,101] and SwinV2-[T,S,B]) from the most widely studied model families, CNNs and Transformers. It is worth pointing out that our approach is able to deliver consistently better accuracy by combining models at different costs while the single best baseline is sensitive to the model cost gaps.
>
> That said, we do appreciate the comment on the single best, and we extend our evaluation to classifiers with small cost gaps (see Table 6, Page 19 in our revision) and show that our approach still consistently outperforms the single best baseline. Results can be found in Table 7, Page 20 in our revision. Specifically, with 40% cost reduction, OCCAM is able to achieve less than 1% accuracy drop while the single best baseline suffers from an accuracy drop of 9.15%.
>
> Furthermore, if the reviewer believes any specific models are required to further reduce the cost gaps and would kindly point out such models, we would be happy to conduct new experiments to address the reviewer’s concern even more thoroughly.
>
> **Q4: Why are the results for Random the same for all cost reduction levels in Table 2 and Table 4, if it solves the same ILP problem as in Equation 4?**
>
> A4: Thank you for this inspiring question! As illustrated in Figure 3 (Page 9), the accuracy achieved by the Random baseline first goes up as the normalized cost budgets (B) increase and quickly plateaus when B exceeds roughly 0.5 according to the plot. In Tables 2 (Page 9) and 4 (Page 17), we report performance of all methods at 10%, 20%, and 40% cost reduction where the Random baseline has plateaued and the performance remains the same. This phenomenon can be explained by the fact that Random estimates the test accuracy for each classifier and test query by uniformly sampling from [0, 1] and solves the ILP problem as in Eq. 4 (see Lines 401-402). According to the random estimates, the best classifier for each test query is also a random variable of a uniform distribution over all available classifiers, whose expected cost is the average cost of all classifiers listed in Table 1 (Page 8), that is, (0.15 + 0.22 + 0.29 + 0.52 + 0.53 + 0.98 + 1) / 7 = 0.527. In other words, the optimal solution of Random can be reached in expectation when the cost budget exceeds 0.527 on average, after which the increase of cost budgets will no longer help the performance, aligning with our observation as discussed above.
>
> Thank you for your time and consideration. We sincerely hope that you find our responses convincing and would consider increasing your rating.

---

> > ### Author Response · Authors · 2024-11-27
> > **Kind Reminder to Reviewer K8Qp**
> >
> > Dear reviewer, this is a gentle reminder that the discussion phase will end in 1 week but we have not yet received your feedback to our rebuttal. We understand that your time is valuable and in high demand due to your other important commitments. But we are eager to work with you to improve this paper, to which we have devoted extensive dedication and effort. We sincerely hope that you find our responses convincing and would consider increasing your rating.

---

> > > ### Comment · Reviewer_K8Qp · 2024-12-01
> > >
> > > Sorry for the late reply due to the Thanksgiving period. Thank you for your detailed answers to my questions. As a follow-up to Q2, what's the percentage used for train-validation split?

---

> > > > ### Author Response · Authors · 2024-12-02
> > > >
> > > > Thank you for your response, and I hope you had a wonderful Thanksgiving!
> > > >
> > > > In our revision, as detailed in Section C.1 (Page 22), we use a 6:4 train-validation split ratio for all datasets where we manually train image classifiers (CIFAR-10, CIFAR-100, and Tiny ImageNet). For ImageNet-1K, which includes 1,281,167 training images and 50,000 validation images, we leverage image classifiers pre-trained on this dataset and uniformly sample 40,000 images from the validation set as our validation data and the remaining ones are treated as the test data.
> > > >
> > > > We sincerely hope that our responses address your concerns and that you would consider increasing your rating.

---

### Official Review · Reviewer_gGnF · 2024-11-02

**Soundness:** 3
**Presentation:** 3
**Contribution:** 3
**Rating:** 6
**Confidence:** 4

**Summary:**

The paper considers minimizing model inference cost by finding the best instance-classifier assignment. A novel method is proposed and there are some new ideas. However, the problem formulation ignores the extra cost induced by the proposed method, which is not discussed in the main text or evaluated in experiments. This is a critical point that might make the proposed method meaningless in some situations.

**Strengths:**

1. The paper presents a new method to find the best instance-classifier assignment to minimize inference cost.
2. The proposed method for accuracy estimation is asymptotically unbiased and the assignment problem can be solved with ILP solvers.
3. Good results are show on a specific evaluation setting.

**Weaknesses:**

1. The problem formulation seems to be unreasonable. The paper aims to find the best instance-classifier assignment to minimize the overall classification inference cost. However, the problem formulation ignores the extra cost for finding the best assignment itself. If the extra cost of finding the best assignment is larger than the saved cost in inference from that assignment, there is no point using the proposed method.
2. The running time and cost of the ILP Solver is not discussed. This is important as it impacts the applicability of the method in practice.
3. The sample size can be as large as 40000, which might presents considerable cost in nearest cost, which however, is also not discussed or reported in experiments.

**Questions:**

1. what is the time complexity and empirical running time of the ILP solver?
2. what is the cost of nearest neighbor search in the sample?
3. Overall, what is the extra cost induced by the method? Would the gain from reduced inference time be significantly larger than the induced extra cost?

---

> ### Author Response · Authors · 2024-11-23
> **To Reviewer gGnF (Part I)**
>
> Thanks for your careful review. We address your concerns below.
>
> Since there are several similar comments on the overhead analysis of our work, we respond to them together for clarity and simplicity.
>
> **Q1: The problem formulation seems to be unreasonable. The paper aims to find the best instance-classifier assignment to minimize the overall classification inference cost. However, the problem formulation ignores the extra cost for finding the best assignment itself. If the extra cost of finding the best assignment is larger than the saved cost in inference from that assignment, there is no point using the proposed method.**
>
> A1: Thank you for the valuable comment. As discussed in Sec 5.1 (Lines 364-392), in practice, the deployment cost (dollars) highly correlates with the GPU usage while the optimal assignment computation (e.g., nearest neighbor search, ILP solving) is carried out by CPUs that are significantly cheaper: e.g., the Azure virtual machine equipped with 1 V100 GPU is 30X more expensive than the CPU-only machines, as discussed in Lines 369-373. In addition to the dollar costs, we have also investigated the latency overheads incurred by optimal assignment computation and found out that the latency overheads are negligible (see our A2 and A3 below). We have included the overhead analysis in Sec A.8, Page 20 in our revision.
>
> **Q2 (Overhead incurred by ILP solver): The running time and cost of the ILP Solver is not discussed. This is important as it impacts the applicability of the method in practice. What is the time complexity and empirical running time of the ILP solver?**
>
> A2: Thank you for the great comment. As discussed in Lines 294-296, Integer Linear Programming (ILP) problem is in general NP-hard; however decades of dedicated effort on efficiently and effectively solving large industry-scale ILP instances has led to ILP solvers that scale to very large instances in practice [1]. In our evaluation, we adopt HiGHS [2] as our ILP solver given its well-demonstrated efficiency and effectiveness on public benchmarks [3]. On the Tiny ImageNet dataset, we investigated the efficiency of the HiGHS ILP solver. With 10,000 test images and 7 classifiers (equivalently an ILP instance with 70,000 variables and constraints, see E.q. 4, Page 6), the HiGHS ILP solvers takes 15.1 seconds to return the optimal assignment. It is worth noting that the latency overhead of ILP solver is only a fraction of the time of using the smallest model (ResNet-18 takes 88.9s to process 10,000 test images, see Table 1, Page 8), and actually less than 2.5% of always using the largest model (SwinV2-B takes 610.6s to process 10,000 test images, see Table 1, Page 8), which demonstrates the overhead incurred by ILP solver is negligible, showing that the problem of finding optimal assignment is well motivated and worth solving.
>
> We note that there are other ILP solvers which are even more efficient than HiGHS. For example, Gurobi [4] is up to 20X faster than HiGHS on large scale MILP instances with up to 640K variables and constraints [5], which can be used as the ILP solver if the problem scale increases further.

---

> ### Author Response · Authors · 2024-11-23
> **To Reviewer gGnF (Part II)**
>
> **Q3 (Overhead incurred by Nearest Neighbor Search): The sample size can be as large as 40000, which might presents considerable cost in nearest cost, which however, is also not discussed or reported in experiments. What is the cost of nearest neighbor search in the sample?**
>
> A3: Thank you for the comment. On Tiny ImageNet, we investigated the overhead incurred by nearest neighbor search. With 10,000 test images and 40,000 total samples, the nearest neighbor search takes 8.68 seconds to return all results, up to two orders of magnitude smaller  than the model inference time (see Table 1, Page 8), which leads to negligible overheads. Such efficiency can be attributed to the linear time complexity of nearest neighbor search w.r.t. the total sample sizes. Specifically, let N denote the number of test images, s denote the sample size, K denote the number of samples, and d denote the image representation dimension; then the time complexity of nearest neighbor search is O(N\*s\*K\*d).
>
>
> **Q4: Overall, what is the extra cost induced by the method? Would the gain from reduced inference time be significantly larger than the induced extra cost?**
>
> A4: As discussed in our A2 and A3, the latency overheads incurred by our method (specifically by ILP solver and nearest neighbor search) are negligible. For example, using the optimal model assignment results in 244 seconds reduction in overall inference time for 10,000 test queries with little to no accuracy drops (see Table 2, Page 9), at 23.8 seconds induced latency overhead. Moreover, our method only requires CPU support to compute the optimal model assignment, incurring significantly small monetary overheads as well.
>
>
> Thank you for your time and consideration. We sincerely hope that you find our responses convincing and would consider increasing your rating.
>
> References:
> [1] Ye, Huigen, Hua Xu, and Hongyan Wang. "Light-MILPopt: Solving Large-scale Mixed Integer Linear Programs with Lightweight Optimizer and Small-scale Training Dataset." The Twelfth International Conference on Learning Representations. 2024.
> [2] https://highs.dev/
> [3] Gleixner, Ambros, et al. "MIPLIB 2017: data-driven compilation of the 6th mixed-integer programming library." Mathematical Programming Computation 13.3 (2021): 443-490.
> [4] https://www.gurobi.com/
> [5] MindOpt Adapter for CPLEX Benchmarking Performance Analysis, 2024

---

> > ### Comment · Reviewer_gGnF · 2024-11-26
> >
> > Thank you for addressing my comments! I have increased the scores.

---

> > > ### Author Response · Authors · 2024-11-27
> > > **Thank you Reviewer gGnF**
> > >
> > > We thank you for recognizing the values of our work and your insightful comments!

---

### Official Review · Reviewer_DVtc · 2024-11-03

**Soundness:** 3
**Presentation:** 3
**Contribution:** 2
**Rating:** 6
**Confidence:** 3

**Summary:**

This paper addresses the trade-off between classification accuracy and inference cost by proposing a framework that combines small and large models. The authors introduce OCCAM (Optimization with Cost Constraints for Accuracy Maximization), a framework that optimally assigns classifiers to queries within a cost budget, leveraging the insight that some “easy” queries can be accurately classified by smaller models. OCCAM uses a statistical accuracy estimator and solves an integer linear programming problem to create a model portfolio that minimizes costs while maintaining accuracy.

**Strengths:**

- The paper demonstrates statistical guarantees to compute optimal assignments with weak assumptions.

- Experimental results on classification datasets show  up to 40% cost reduction with
no significant drop in classification accuracy.

**Weaknesses:**

- Limited scope of experiments with pretrained classifiers:
The paper's most significant weakness is the limited number of pretrained classifiers used in the experiments. This scope may not provide a comprehensive evaluation of the proposed method and could affect the generalizability of the results.


- Insufficient clarification on related work:
The paper lacks adequate clarification on how it builds upon or differentiates from existing research in the field. Providing a clearer context within the related work section (as highlighted in the Questions) would strengthen the paper by situating it more effectively within the current academic discourse.

**Questions:**

- How does OCCAM fit into the literature of routing for Mixture-of-experts?

- What is the reason for K = 40 being the maximal value?

- How does OCCAM perform in scenarios with more than 7 classifiers? E.g. model selection literature [1] use more than 100 pretrained models for ImageNet.



[1] Mohammad Reza Karimi, Nezihe Merve Gürel, Bojan Karlaš, Johannes Rausch, Ce Zhang, and Andreas Krause. Online active model selection for pre-trained classifiers. In International Conference on Artificial Intelligence and Statistics (AISTATS), pp. 307–315. PMLR, April 2021.

---

> ### Author Response · Authors · 2024-11-23
> **To Reviewer DVtc (Part I)**
>
> Thanks for your careful review. We address your concerns below.
>
> **Q1: Limited scope of experiments with pretrained classifiers: The paper's most significant weakness is the limited number of pretrained classifiers used in the experiments. This scope may not provide a comprehensive evaluation of the proposed method and could affect the generalizability of the results.**
>
> A1. Thank you for the comment. Our current evaluation covers 7 mainstream image classifiers (ResNet-[18,34,50,101] and SwinV2-[T,S,B]) from the most widely studied model families, CNNs and Transformers. It is worth noting that recent papers [2,3,4] in hybrid ML inference typically include evaluation with 2 or 4 models, which we outdo in our evaluation. Nevertheless, taking this comment into account, we undertake an evaluation of OCCAM in the presence of a large number of classifiers: specifically, we conduct experiments with more than 100 classifiers on ImageNet. Our results (see Sec A.6, Page 17 in our revised manuscript) demonstrate the effectiveness of OCCAM. Specifically, with 40% cost reduction, OCCAM is able to achieve less than 1% accuracy drop while the baselines suffer from accuracy drops of at least 7%.
>
> **Q2: Insufficient clarification on related work: The paper lacks adequate clarification on how it builds upon or differentiates from existing research in the field. Providing a clearer context within the related work section (as highlighted in the Questions) would strengthen the paper by situating it more effectively within the current academic discourse.**
>
> A2: Thank you for the comment. In Sec 2, we discussed the related work from three domains –  efficient ML inference, hybrid ML inference, and image classification, for which we clarify the relationship and difference with OCCAM. We have included discussion (see our A3 and A5 below) on the references that the reviewer pointed out to better position our work.
>
> **Q3: How does OCCAM fit into the literature of routing for Mixture-of-experts?**
>
> A3: Thank you for this valuable comment. Both OCCAM and mixture-of-experts (MoE) involve dynamically deciding which model (or "expert") processes a given query based on specific criteria. However, MoE often assumes homogenous experts (e.g., In [5], experts are identical feed-forward networks within the transformer layers), with routing decisions made by learned gating functions that require significant computational resources to re-train if we want to add/update/delete experts. In contrast, OCCAM is training-free and can leverage diverse pre-trained models (e.g., CNNs and Transformers) and provides principled routing decisions with statistical guarantees, making it more practical for scenarios where continual re-training is infeasible, and cost constraints are critical.
>
> Last but not the least, MoE as a framework is orthogonal to OCCAM. Given that MoE often results in a single sparsely-activated model, it can be regarded as an input for OCCAM. An interesting line of future work would be to combine MoE with OCCAM to achieve further cost reduction. We have added the discussion in Sec 2, Page 3 of our revised manuscript.

---

> ### Author Response · Authors · 2024-11-23
> **To Reviewer DVtc (Part II)**
>
> **Q4: What is the reason for K = 40 being the maximal value?**
>
> A4. As shown by Lemma 4.5 and 4.6, our estimator is asymptotically unbiased and low-variance as both s (sample size) and K (number of samples) increase, which leads to better overall performance. We demonstrate this by increasing K from 10 to 40 while keeping s fixed (1000), and increasing s from 400 to 1,000 while keeping K fixed (40), as illustrated in Figure 13(a-b), Page 20 in our revision. We observe that the performance of OCCAM consistently dominates that of FrugalMCT and improves as both K and s increase. In our evaluation, we pre-compute a held-out dataset (e.g., for Tiny ImageNet, we uniformly sample 40,000 images from the 100,000 training images) from which we draw K samples of size s. With the maximal number of samples bounded (for Tiny ImageNet, K*s <= 40,000), there is a trade-off between increasing K and having larger s. As shown in Figure 13(c), Page 20 in our revision, though a larger K typically leads to better accuracy, it also limits the maximal value that s can take (s <= 40,000 / K), which results in sub-optimal performance. We empirically choose K=40 since it gives us the best overall accuracy.
>
> **Q5: How does OCCAM perform in scenarios with more than 7 classifiers? E.g. model selection literature [1] use more than 100 pretrained models for ImageNet.**
>
> A5. Thank you for the insightful question. [1] considers an online setting where the model performance is unknown and the primary goal is to identify the best-performing model using minimal labeled data, emphasizing selective sampling strategies. In contrast, OCCAM focuses on achieving the highest accuracy across all queries while minimizing total inference cost, leveraging cost-aware classifier assignment.
>
> Nevertheless, we have taken this comment into account and conducted an evaluation of OCCAM in the presence of a large number of classifiers: specifically, we conducted experiments with more than 100 classifiers on ImageNet to show the effectiveness of OCCAM. The results (see Sec A.6, Page 17 in our revision), as we have also discussed in our A1, attest to the effectiveness of OCCAM in this setting.
>
> Thank you for your time and consideration. We sincerely hope that you find our responses convincing  and would consider increasing your rating.
>
> References:
> [1] Mohammad Reza Karimi, Nezihe Merve Gürel, Bojan Karlaš, Johannes Rausch, Ce Zhang, and Andreas Krause. Online active model selection for pre-trained classifiers. In International Conference on Artificial Intelligence and Statistics (AISTATS), pp. 307–315. PMLR, April 2021.
> [2] Kag, Anil, and Igor Fedorov. "Efficient edge inference by selective query." International Conference on Learning Representations. 2023.
> [3] Ding, Dujian, et al. "Hybrid LLM: Cost-Efficient and Quality-Aware Query Routing." International Conference on Learning Representations. 2024
> [4] Chen, Lingjiao, Matei Zaharia, and James Zou. "Efficient online ml api selection for multi-label classification tasks." International conference on machine learning. PMLR, 2022.
> [5] Fedus, William, Barret Zoph, and Noam Shazeer. "Switch transformers: Scaling to trillion parameter models with simple and efficient sparsity." Journal of Machine Learning Research 23.120 (2022): 1-39.

---

> > ### Comment · Reviewer_DVtc · 2024-11-26
> > **Clarification Acknowledgement**
> >
> > Thank you for your clarifications. I have changed my score.

---

> > > ### Author Response · Authors · 2024-11-27
> > > **Thank you Reviewer DVtc**
> > >
> > > Thank you for the recognition of our work and all the thoughtful reviews!

---

### Author Response · Authors · 2024-11-23
**General Response**

We thank all the reviewers for their careful reviews and excellent comments. We have added a range of new experiments which address the concerns raised by the reviewers and we believe this has made the evaluation of our approach much more comprehensive. The changes are highlighted in blue in the updated draft and are summarized below:

1. We have discussed the related work pointed out by Reviewer DVtc and Reviewer iJX4 in Section 2 (Page 3).
2. We have shown that the computational overhead of OCCAM is negligible in Appendix A.8 (Page 20) as suggested by Reviewer gGnF and Reviewer iJX4.
3. We have conducted additional experiments to show how to apply OCCAM to more than 100 pre-trained classifiers, as suggested by Reviewer DVtc. These experiments are presented in Appendix A.6 (Page 17) and demonstrate the effectiveness of OCCAM. Specifically, with 40% cost reduction, OCCAM is able to achieve less than 1% accuracy drop while the baselines suffer from accuracy drops of 7% or more.
4. We have added experiments to clarify how to effectively choose K (number of samples) and s (sample size) in practice using a held-out set in Appendix A.7 (Page 18) as suggested by Reviewer DVtc.
5. We have included new baselines to present the upper bound performance that OCCAM can reach with perfect accuracy estimator in Appendix A.9 (Page 21) as suggested by Reviewer iJX4;
6. We have performed analysis on the complementarity between different classifiers in Appendix A.10 (Page 21) as suggested by Reviewer iJX4;

The details of revision are referred to the following official comments.

---

### Public Comment · ~Janek_Haberer1 · 2025-02-11
**Question about Feature Extractor**

Congratulations on the acceptance of your paper!

I had a question regarding the step “Accuracy Estimation Using Nearest Neighbors”. Are you doing the nearest neighbor search based on the raw input image or on a feature representation of the image? In the supplementary material, you have multiple figures showing the impact of different feature extractors, suggesting that you are using a network such as ResNet18/ResNet50/SwinV2-T as a feature extractor. However, this would cause extra cost as you can’t reuse the feature representations when routing to a different network, right?

I would appreciate it if you could clarify this part!

---

> ### Public Comment · ~Dujian_Ding1 · 2025-02-11
>
> Hi Janek,
>
> Thank you for the question!
>
> Yes, as clarified in Lines 407-409, we extract the feature representation to compute nearest neighbours. The incurred costs of feature extraction have been "deducted from the user budget B before we compute the optimal model portfolio" to ensure the overall cost is within given budget. Notably, the feature representation only needs to be computed once per image and can be re-used for nearest neighbour search when routing to different models. In this paper, we develop an end-to-end solution for classification queries which assumes a fixed set of ML classifiers and takes raw image as inputs. One alternative is to pre-compute the feature representation and compute the optimal model portfolio for different model setups, which may be worth future research.
>
> Please let me know if you have any further questions.
>
>
> Kind Regards,
> Dujian

---

> > ### Public Comment · ~Janek_Haberer1 · 2025-02-12
> >
> > Ah, I somehow must have missed that part, thanks for the clarification!

---

### Meta-Review · Area_Chair_teU3 · 2024-12-20

**Metareview:**

This paper proposes OCCAM, a principled approach to optimize classifier assignments across queries, maximizing accuracy under user-specified cost budgets by solving an integer linear programming problem. Experiments demonstrate that OCCAM reduces inference costs by up to 40% with minimal accuracy loss on real-world datasets. The overall review of the paper is positive, so the paper is recommended for acceptance at this time.

**Additional Comments On Reviewer Discussion:**

During the rebuttal period, the authors addressed the reviewers' concerns, and as a result, reviewers have increased the score.

---

### Decision · Program_Chairs · 2025-01-22

Accept (Poster)